# The bacterial quorum sensing signal DSF hijacks *Arabidopsis thaliana* sterol biosynthesis to suppress plant innate immunity

Tuan Minh Tran[1],*, Zhiming Ma[1],*, Alexander Triebl[2],†, Sangeeta Nath[3,4],†, Yingying Cheng[5], Ben-Qiang Gong[6], Xiao Han[1], Junqi Wang[7], Jian-Feng Li[6], Markus R Wenk[2], Federico Torta[2], Satyajit Mayor[3,8], Liang Yang[5,9], Yansong Miao[1,10]

**Quorum sensing (QS) is a recognized phenomenon that is crucial for regulating population-related behaviors in bacteria. However, the direct specific effect of QS molecules on host biology is largely understudied. In this work, we show that the QS molecule DSF (*cis*-11-methyl-dodecenoic acid) produced by *Xanthomonas campestris* pv. *campestris* can suppress pathogen-associated molecular pattern–triggered immunity (PTI) in *Arabidopsis thaliana*, mediated by flagellin-induced activation of flagellin receptor FLS2. The DSF-mediated attenuation of innate immunity results from the alteration of FLS2 nanoclusters and endocytic internalization of plasma membrane FLS2. DSF altered the lipid profile of *Arabidopsis*, with a particular increase in the phytosterol species, which impairs the general endocytosis pathway mediated by clathrin and FLS2 nano-clustering on the plasma membrane. The DSF effect on receptor dynamics and host immune responses could be entirely reversed by sterol removal. Together, our results highlighted the importance of sterol homeostasis to plasma membrane organization and demonstrate a novel mechanism by which pathogenic bacteria use their communicating molecule to manipulate pathogen-associated molecular pattern–triggered host immunity.**

## Introduction

Bacteria use quorum sensing (QS) to precisely coordinate population behaviors in response to various environmental cues. QS signals are small molecules that contribute to bacterial virulence in bacterial–host interactions by either regulating bacterial type III secretion or priming host immune systems (Newman et al, 2004; Chatterjee et al, 2008; Brock et al, 2010). Although the contribution of QS signals to bacterial pathogenicity has been examined extensively, the direct effects of these small molecules on host biology remain under-explored.

QS signals produced by human pathogens, such as N-3-oxo-dodecanoyl homoserine lactone (3OC12-HSL), modulate pathogen-associated molecular pattern (PAMP)–mediated NF-kB signaling in macrophages and innate immune responses in recognition of LPS—thereby promoting persistent infection (Kravchenko et al, 2008). The 3OC12-HSL was also shown to be recognized by the bitter taste receptor, T2R family protein, T2R38 in neutrophils, lymphocytes, and monocytes to activate innate immune responses (Maurer et al, 2015). Similarly, plant pathogenic bacteria also secrete small-molecule virulence factors to cross-talk with host and manipulate host immunity during infection. QS molecules produced by several bacterial phyto-pathogens showed a wide range of interference in host development and defense mechanisms, including the root morphological development in an auxin-dependent manner or callose deposition during immune responses (Schikora et al, 2011; Bai et al, 2012; Zhao et al, 2015). However, the detailed mechanisms of how QS molecules directly influence plant host development and pathology, especially the PAMP-mediated host immunity, remains elusive.

Here, we studied the host–pathogen communication between host *Arabidopsis thaliana* and a specific QS molecule, the diffusible signal factor (DSF). The DSF QS molecule family is produced by diverse Gram-negative bacteria, many of which are devastating phytopathogens such as *Xanthomonas campestris* pv. *campestris*

---

[1]School of Biological Sciences, Nanyang Technological University, Singapore, Singapore    [2]Department of Biochemistry, Singapore Lipidomics Incubator (SLING), Yoo Loo Lin School of Medicine, National University of Singapore, Singapore, Singapore    [3]Institute for Stem Cell Biology and Regenerative Medicine, Bangalore, India    [4]Manipal Institute of Regenerative Medicine, Manipal Academy of Higher Education, Bangalore, India    [5]Singapore Centre for Environmental Life Sciences Engineering, Nanyang Technological University, Singapore, Singapore    [6]School of Life Sciences, Sun Yat-sen University, Guangzhou, China    [7]Department of Biology, Southern University of Science and Technology, Shenzhen, China    [8]National Centre for Biological Sciences, Tata Institute for Fundamental Research, Bangalore, India    [9]School of Medicine, Southern University of Science and Technology, Shenzhen, China    [10]School of Chemical and Biomedical Engineering, Nanyang Technological University, Singapore, Singapore

Correspondence: yansongm@ntu.edu.sg
Alexander Triebl's present address is Syngenta, Münchwilen AG, Canton of Aargau, Switzerland
*Tuan Minh Tran and Zhiming Ma contributed equally to this work
†Alexander Triebl and Sangeeta Nath contributed equally to this work

---

(*Xcc*), *Xanthomonas oryzae* pv. *oryzae*, *Xylella fastidiosa*, as well as several human pathogens, including *Pseudomonas aeruginosa*, *Stenotrophomonas maltophilia*, and *Burkholderia cenocepacia* (Zhou et al, 2017). Notably, *X. campestris* pv. *campestris* (*Xcc*), a global-threat phytopathogen that causes black rot on crucifers (Meyer et al, 2005), produces up to four different analogs of DSF (*cis*-2 unsaturated fatty acids), with DSF (*cis*-11-methyl-dodecenoic acid) being the primary compound (>75%) at concentrations of 40–100 *μ*M during infection (Kakkar et al, 2015). DSF regulates the expression of *Xcc* type III secretion system, host cell wall–macerating enzymes and other traits in *Xcc* (Zhou et al, 2015, 2017), and elicited host immune response with an enhanced *Arabidopsis PR-1* gene expression (Kakkar et al, 2015). However, the mechanism by which this QS molecule modulates plant defense responses is mostly unknown.

In this study, we investigated the mechanisms by which DSF alters host physiology and pathology. We demonstrated that DSF induces multiple macro- and microscopic changes in *A. thaliana* cell biology and development, including root morphogenesis and plant lipid profiles. The DSF-induced phytosterol production that impaired both clathrin-mediated endocytosis (CME) for internalizing FLS2 and FLS2 nano-clustering on the cell surface, which therefore desensitized *Arabidopsis* immune responses to the bacterial flagellum. Removing the DSF-induced sterol accumulation in *Arabidopsis* could reverse the defects in host endocytosis, development, and immunity caused by DSF. However, the FLS2–BAK1 interaction and MAPK activation by flg22 elicitation were unchanged, indicating uncoupled PTI responses underlying DSF immune signaling. Our studies have significant implications for the understanding of bacterial QS molecule DSF function in host biology and host pattern-triggered immunity (PTI) pathways.

# Results

## DSF suppresses bacterial flagellin-triggered innate immune responses of *Arabidopsis*

Plants have developed multiple sophisticated defense mechanisms to recognize and respond to a wide range of bacterial virulence factors. Moreover, bacterial pathogens strategically instigate or subvert the host immune response by interfering with PAMP recognition (He et al, 2006; Zhang et al, 2007). We asked whether DSF, a recently discovered QS signal produced by diverse Gram-negative pathogens (Ryan et al, 2015), could dysregulate plant growth and PAMP-triggered immunity (PTI) responses. We first examined *Arabidopsis* growth in response to DSF molecules within the pathological concentration of DSF during infection (Kakkar et al, 2015) and found that DSF inhibited primary root growth of *A. thaliana* in a dose-dependent manner without apparent effects on seed germination (Fig S1A). In addition, to test whether DSF influences flagellin-triggered immunity, we tested several flagellin-triggered *Arabidopsis* responses in plants treated with 25 *μ*M of DSF, the concentration at which root growth was inhibited by 40%. We found that flg22-induced stomatal closure (aperture reduced from 4 to 2 *μ*m) was impaired by DSF treatment (Fig 1A). At low concentrations, DSF itself did not trigger callose deposition (Fig S1B), in

agreement with a previous report (Kakkar et al, 2015). However, we found that callose deposition, a plant physical defense responses to prevent pathogens infections (Gómez-Gómez & Boller, 2000), was significantly impaired in seedlings treated with DSF before flg22 elicitation (Fig 1B), as was flg22-induced reactive oxygen species (ROS) burst by half (Fig 1C). This inhibition of ROS by DSF was dose-dependent and not only limited to ROS induced by flg22 but also effective on ROS burst induced by the PAMP elf26 (Fig S1C and D). We next performed a plant infection assay, in which *Arabidopsis* plants pretreated with DSF or DMSO (mock) were inoculated with *Pseudomonas syringae* pv. *tomato* DC3000 (*Pst* DC3000). DSF does not induce responses from bio-reporter strains that detect acyl homoserine lactones (AHLs), the QS signals used by *Pseudomonas spp.* (Barber et al, 1997; Elasri et al, 2001; Quiñones et al, 2004). The use of a non-DSF–producing bacterium here (*Pst* DC3000) was to differentiate the direct effect of DSF on plant immunity versus the effect of DSF through the activation of the bacterial type III secretion system and other traits regulated by QS in *Xcc*, which would otherwise be difficult to dissect as DSF-deficient or overproducing mutants are less virulent on plants (Torres et al, 2007; Gudesblat et al, 2009).

In this assay which involved using live bacteria with all the potential PAMPs, we found that DSF-treated plants were colonized by a higher number of *Pst* cells than control plants, indicating a higher susceptibility of plants to *Pst* infection in the presence of DSF (Fig 1D). We further examine the ability of this QS molecule in compromising the defense mechanism of the plants that were pre-protected by PTI-signaling. In treatments where plants were primed with flg22 peptide before bacterial inoculation, we found that DSF application before flg22 priming significantly lowered flg22-induced protection against infection, reflected in a higher bacterial population in DSF+flg22 treatment than that of DMSO+flg22 treatment (Fig 1D and E).

## DSF interferes with the host endocytic internalization on the plasma membrane (PM) without compromising MAPK signaling

We next sought to investigate if DSF could interfere with the dynamics and functions of the *Arabidopsis* flagellin receptor FLS2 (Gómez-Gómez & Boller, 2000). Flagellin binding causes the rapid association of the FLS2 receptor with its interacting partners to activate the innate immunity cascade and subsequently leads to the endocytic internalization of the receptor (Robatzek et al, 2006; Lu et al, 2010; Beck et al, 2012).

We first tested whether DSF affects FLS2 endocytosis upon elicitation with its corresponding ligand flg22. Upon treatment of flagellin peptide flg22 but not flgII-28 (a flagellin epitope that *Arabidopsis* does not recognize, as a negative control), FLS2 endocytic internalization was evidenced by the appearance of punctate endosomes after around 60–75 min and disappeared at 120 min post-elicitation (Figs 2A and S1E) (Robatzek et al, 2006; Hind et al, 2016; Mbengue et al, 2016). In contrast, when FLS2-GFP seedlings were treated with DSF for 24 h before flg22 exposure, we observed an apparent delay in FLS2 internalization upon PAMP elicitation. After 60 min of flg22 elicitation, DSF-treated plants showed fewer FLS2-positive endosomes than DMSO-treated plants, indicating attenuated endocytosis of FLS2. In addition, we observed

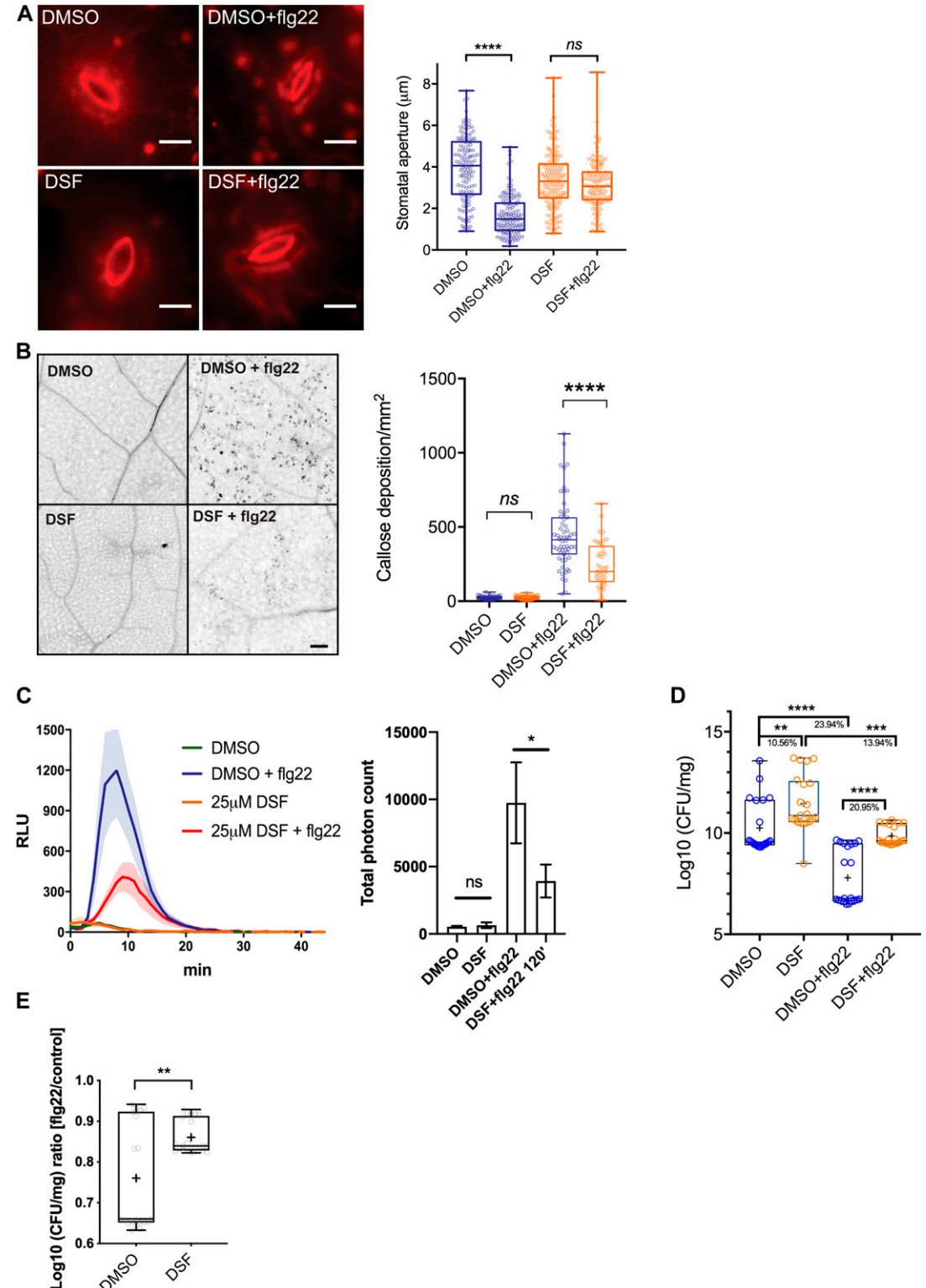

**Figure 1. Diffusible signal factor (DSF) simultaneously dampens several plant defense responses in *Arabidopsis thaliana* stimulated by the bacterial pathogen-associated molecular pattern flg22.**
**(A)** Stomatal apertures of 5-wk-old Col-0 leaves after flg22 elicitation. Intact leaves were infiltrated with DMSO or 25 $\mu$M DSF diluted in 10 mM MgCl$_2$ for 24 h before the epidermal layers of detached leaves were peeled off and treated with 1 $\mu$M flg22, stained with propidium iodine, and imaged. Stomatal apertures were measure by Fiji (n ≥ 130 stomata from n = 5 plants; bars, 10 $\mu$m). The experiment was repeated twice with a similar result. **(B)** Callose deposition in 2-wk-old Col-0 plants pretreated with DMSO or 25 $\mu$M DSF in 1/2 liquid MS for 24 h before being elicited by flg22 in liquid 1/2 MS. Leaves were stained with aniline blue and imaged using a confocal microscope

that this inhibition of FLS2 endocytosis was time-dependent as longer incubation with DSF further reduced the number of FLS2-positive endosomes at 60 min post-elicitation with flg22 (Fig S1F). We then asked whether the attenuated FLS2 endocytosis could also lead to a delay in FLS2 degradation (Scheuring et al, 2012; Ben Khaled et al, 2015). Consistent with a previous observation (Smith et al, 2014), we found that FLS2 degradation occurred ~30 min after peptide elicitation, and this response was delayed in DSF pretreated seedlings (Fig S2A).

To further confirm the DSF-induced defects in receptor endocytosis, we examined FLS2 endocytic internalization at the PM using variable-angle epifluorescence microscopy (VAEM) (Fig 2B). From VAEM micrographs, we quantitatively determined receptor internalization by kymograph analysis of FLS2-GFP puncta. We observed that flg22 enhanced FLS2 endocytosis as shown by the shortened lifetime of FLS2 on the PM, and this enhancement was blocked by DSF pretreatment before peptide elicitation (Fig 2C). As FLS2 is endocytosed via CME (Mbengue et al, 2016), we then tested whether the delay in FLS2 endocytosis was due to a general inhibition of CME. Indeed, we observed a significant increase in the lifetime of clathrin light chain (Fig S2E), as well as the lifetime of several typical CME-dependent endocytic receptors: brassinosteroid receptor BRI1 and boron receptor BOR1 under DSF treatment (Fig S2F and G), indicating the impaired endocytic internalization of these PM receptors.

We did not observe a noticeable change in the acute response of MAPK phosphorylation, which occurs ~15 min post-elicitation and returns to the basal level within 60 min (Fig S2B). This result suggests that flg22-induced MAPK activation is independent of FLS2 endocytosis under DSF treatment, which is consistent with the unchanged *FRK1* up-regulation in the presence of flg22 (Fig S2C) that is through the activation of an MAPK signaling cascade (Asai et al, 2002). Exposure to flg22 could still induce the expression of *FRK1* in DSF-treated seedlings, suggesting that DSF did not have a significant effect on the flg22-induced accumulation of *FRK1* transcript and might not influence FLS2–BAK1 heterodimerization that was critical for *FRK1* induction (Sun et al, 2013). In agreement with this observation, we also found that DSF did not affect the flg22-induced FLS2-BAK1 heterodimerization (Fig S2D), suggesting that the inhibition of flg22-induced ROS burst by DSF is unlikely a direct consequence of disturbed FLS2–BAK1 association, and uncoupled from MAPK activation.

## DSF impairs lateral nano-clustering of FLS2 on the PM

In addition to the endocytic internalization that removes material from the PM, the lateral nano-clustering of PM-bound receptors through multivalent interactions is well known to mediate diverse signal transduction mechanisms during host–pathogen interactions

(Bücherl et al, 2017; Liang et al, 2018). The self- or hetero-oligomerization of surface receptors are critical for receptor activation, signal amplification, and signal transduction in both plants and humans, such as the brassinosteroid receptor BRI1 (Wang et al, 2005), chitin receptor *At*CERK1 (Liu et al, 2012), EGF receptor (Schlessinger, 2002; Endres et al, 2013), or insulin receptor IR/IGFR (Cabail et al, 2015). To further investigate DSF-induced effects on receptor lateral dynamics and their activity, we performed living cell imaging of FLS2–GFP, chosen from several DSF-hindered endocytic PM receptors (Figs 2C and S2F and G). FLS2 formed heterogeneous PM clusters with or without ligand activation (Fig 2B), representing the resting- or activated-states, respectively. FLS2–FLS2 association was also reported, although this association can occur without a ligand (Sun et al, 2012). The formation of the FLS2–BAK1 heterodimer upon flg22 elicitation is a key step in activating downstream defense signaling (Chinchilla et al, 2007; Roux et al, 2011). Although DSF did not influence flg22-triggered FLS2–BAK1 interaction and MAPK activation (Fig S2B and D), the fact that DSF impaired the ROS production suggested uncoupled mechanisms in the activation of MAPK and the ROS production by differential inter- or intra-protein interactions of FLS2 for these processes. Consistently, with a DSF pretreatment, FLS2 clusters became more diffused with a lower spatial-clustering index (SCI) after 60 min of flg22 elicitation (Fig 2D).

To study the detailed physical interactions of FLS2 receptors in the multivalent nanoclusters, we took advantage of a well-established living cell imaging approaches in studying mammalian receptor clustering and activation (Ghosh et al, 2012; Kalappurakkal et al, 2019). We quantitatively measured the lateral clusters of FLS2 at nano-scale on the PM directly under ligand activation by performing FRET between identical fluorophores (homo-FRET) coupled with VAEM, which allows sensitive measurement of fast cargo-induced receptor activation (Saha et al, 2015). In this method, abundance of FLS2–GFP clusters at nano-scale will be accompanied by a reduction of fluorescence emission anisotropy of FLS2–GFP when it is excited by polarized light. We determined the steady-state fluorescence emission fluorescence anisotropy of FLS2–GFP with or without 5 min of peptide elicitation (Figs 2E and S2H–K). First, we observed that the anisotropy values of FLS2 receptor foci are independent of their signal intensity (Fig S2J). Flg22 stimulated a significant decrease in anisotropy of FLS2–GFP, reflecting the flg22-induced nano-clusters of FLS2 (Figs 2E and S2K). However, DSF-treated plants showed a reduction of flg22-triggered anisotropy change, indicating a suppressed formation of nano-clusters (Cui et al, 2018). Seedlings exposed to DSF did not exhibit any significant change in emission anisotropy of FLS2 after flg22 elicitation (compared with DMSO+flg22 control, $P > 0.05$) (Figs 2E and S2K), consistent with SCI analysis (Fig 2D).

We next investigated the effects of DSF on bulk endocytosis using the lipophilic dye FM4-64 and CME using transgenic plant

with UV excitation to visualize flg22-triggered callose deposition. Plants treated with only DMSO or DSF were used as negative control (n ≥ 40 images from n = 8 plants; bar, 100 μm). The experiment was repeated twice. **(C)** Reactive oxygen species production and total photon count of reactive oxygen species burst in Col-0 leave strips pretreated with DMSO or 25 μM DSF for 24 h before flg22 elicitation (n ≥ 6 leaf discs/condition). The experiment was repeated three times. **(D)** Bacterial population at 4 d postinoculation in *Arabidopsis* 2-wk-old seedlings pretreated with DMSO (solvent control) or DSF, followed by priming with flg22 peptide and flooding assay with *Pst* DC3000. Boxplot represent pooled data from two independent experiments (n = 21). **(E)** Relative ratio of $\log_{10}$ (CFU/mg) was calculated by dividing the bacterial population of flg22-treated group and the control group for each of the treatment (DMSO or DSF). *P*-values were determined by one-way ANOVA (\*$P < 0.05$; \*\*$P < 0.01$; \*\*\*$P < 0.001$; \*\*\*\*$P < 0.0001$; ns, not significant).

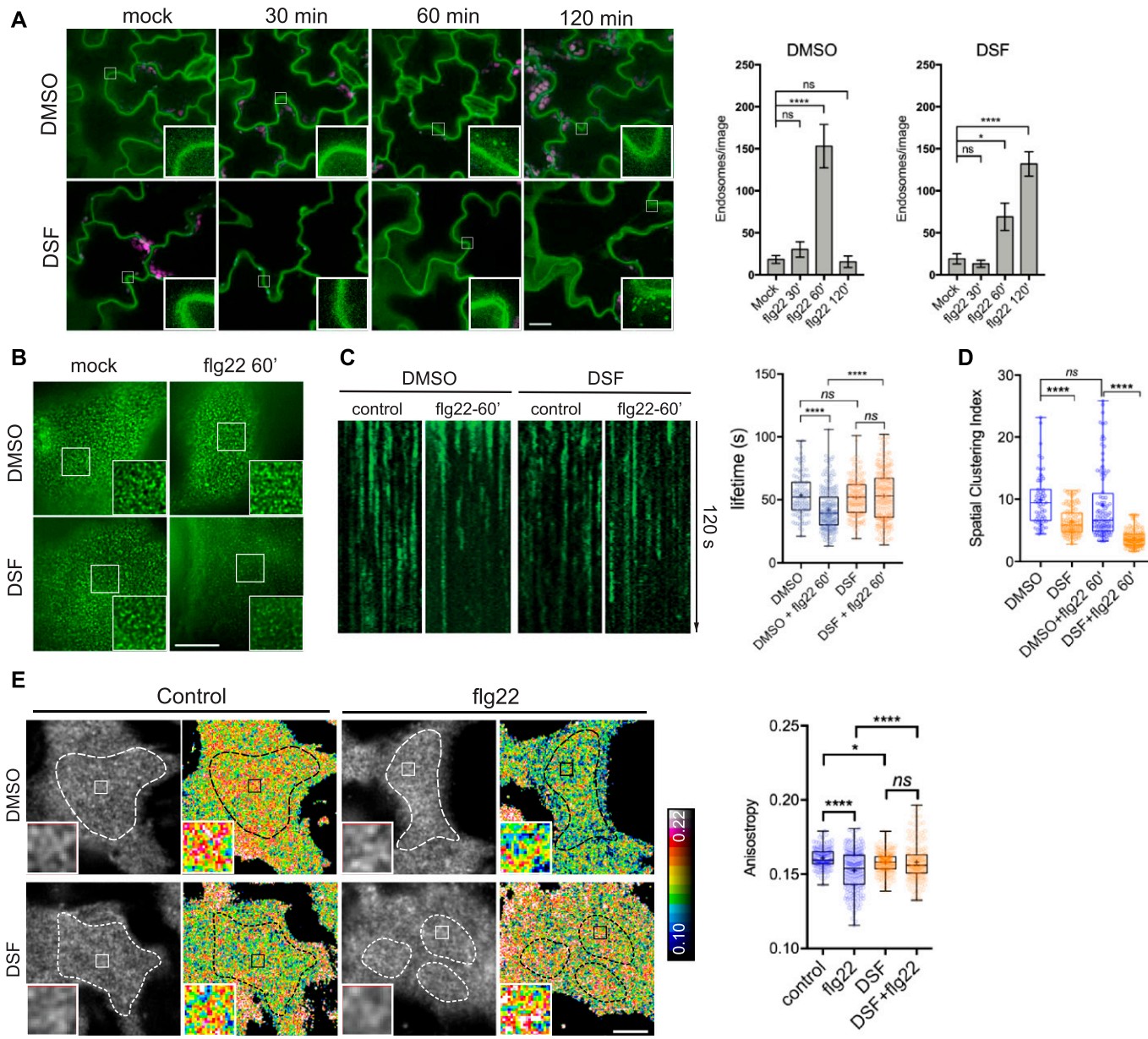

**Figure 2. Diffusible signal factor (DSF) delays ligand-induced endocytosis of the FLS2 receptor in response to flg22 peptide.**
**(A)** 5-d-old FLS2–GFP seedlings were treated for 24 h in liquid 1/2 MS medium supplemented with 25 $\mu$M DSF or DMSO (solvent control) before being elicited with 10 $\mu$M flg22 and imaged at indicated time points. Micrographs represent maximum projections of 12 slices every 1 $\mu$m z-distance (bars, 20 $\mu$m). Bar graphs represent the numbers of endosomes per image. Error bars, SEM (n ≥ 12 from three plants). The experiment was repeated twice with a similar result. **(B)** FLS2–GFP receptors on the plasma membrane (PM) of *Arabidopsis* cotyledons. 5-d-old FLS2–GFP plants were treated with DMSO or DSF for 24 h before elicitation by 10 $\mu$M flg22 and imaged using VA-TIRFM (bar, 1 $\mu$m). **(C)** Kymographs of FLS2-GFP clusters on *Arabidopsis* PM before and after 60 min of flg22 elicitation, following DMSO or 25 $\mu$M DSF 24-h pretreatment. The lifetime of FLS2–GFP foci was measured from kymographs using Fiji (n ≥ 100 endocytic events from n ≥ 6 individual cells). **(D)** Boxplot represents spatial clustering of FLS2 receptor foci, indicated by the spatial clustering index. Spatial clustering index was calculated from TIRF time-lapse movies based on the ratio of top 5% of pixels with the highest intensity and 5% of pixels with the lowest intensity (n ≥ 60 ROIs from n ≥ 6). The experiment was performed twice. **(E)** Homo-FRET analysis of FLS2 receptor clustering on PM of DMSO- and DSF-treated FLS2–GFP seedlings upon flg22 elicitation for 5 min. As the PM is not definitely flat, anisotropy quantification was performed only in those well-focused regions of the PM (white dashed lines) (bar, 5 $\mu$m). Boxplot represents anisotropy values calculated from homo-FRET imaging of FLS2–GFP. Representative intensity and anisotropy images are shown. Insets are representative 20 × 20 pixel ROIs used for data analysis (n ≥ 150 ROIs from n ≥ 8 cells taken from three seedlings). The homo-FRET experiments were performed twice. *P*-values were determined by one-way ANOVA (**P* < 0.05; ***P* < 0.01; ****P* < 0.001; *****P* < 0.0001; ns, not significant).

expressing PIN2–GFP (Xu et al, 2006). Briefly, we monitored the uptake of FM4-64 in *Arabidopsis* seedlings expressing PIN2–GFP, an auxin efflux carrier protein and a CME-dependent cargo (Robert

et al, 2010) in the presence of the novel small-molecule inhibitor ES9-17, which specifically binds to clathrin heavy-chain without protonophoric effect (Dejonghe et al, 2016, 2019). To test whether

DSF would influence CME further on top of a partially compromised CME by small molecular inhibitors (Miao et al, 2006), we pretreated PIN2–GFP seedlings with DMSO/DSF and then performed pharmacological studies with different combinations of ES9-17/BFA on treated plants, using FM4-64 dye as an indicator of bulk endocytosis. Our result indicated that whereas neither DSF nor 50 $\mu$M ES9-17 completely blocked the internalization of FM4-64 dye, a combined treatment of DSF and ES9-17 totally inhibited FM4-64 uptake, indicated by the absence of intracellular FM4-64 signal and FM4-64-containing BFA bodies in BFA treatment (Fig 3A–D). Our results demonstrated the additional attenuation of bulk-phase endocytosis by DSF on top of the CME inhibitor ES9-17.

We then asked whether such impairment of plant CME could also disrupt endosomal transport after endocytic invagination from the PM. We examined the intracellular membrane compartments of the protein sorting and endosomal transport pathways upon DSF treatment. No noticeable ultrastructural changes were observed in early endosome/trans-Golgi network (VHA-a1), late endosome/multivesicular body (MVB) (RabF2b), and Golgi (Got1p homolog), nor in their response to Brefeldin A (BFA) or Wortmannin that induce Golgi aggregation or MVB dilation (Geldner et al, 2009), respectively (Fig S3A–D). The above results suggest that DSF likely impairs plant endocytosis mainly at the step of endocytic internalization on the PM and not at the endomembrane trafficking steps.

### DSF increased host phytosterol contents and reprogramed Arabidopsis lipid metabolism

Dynamic behavior of PM proteins and biological processes in plants, including the endocytic invagination, lateral diffusion, and intermolecular interactions, etc. rely on the lipid-compartmentalization and a balanced sterol composition of the PM (Grison et al, 2015; Gronnier et al, 2017; Kalappurakkal et al, 2019; Platre et al, 2019). We next sought to determine if the lipid profile of Arabidopsis was also altered by DSF, given that this QS signal is a short-chain fatty acid–like molecule that could serve as a building block to create diverse biologically active molecules (Chatterjee et al, 2008). Our targeted lipidomic profiling of Arabidopsis seedlings revealed that Arabidopsis seedlings grown on DSF-supplemented medium had a comprehensive shift in a wide range of lipid compounds compared with those grown on control medium (Fig 3E and F and Supplemental Data 1 and 2). Interestingly, the most notable changes were observed in several acyl steryl glycosides, of which the most abundant phytosterols—sitosterol, campesterol, and stigmasterol showed an increase by two to fourfold. Sterols regulate a wide range of cellular processes, such as lipid metabolism and cell membrane dynamics, such as the formation of membrane nanodomains (Laloi et al, 2007; Wollam & Antebi, 2011). We, therefore, tested whether the increased sterol content could explain the DSF-induced phenotypes by treating Arabidopsis seedlings with Methyl-$\beta$-cyclodextrin (M$\beta$CD) in combination with DSF. M$\beta$CD is a widely used sterol-depleting agent that acts strictly at the membrane surface by binding to sterol with high affinity and remove sterol acutely within minutes, with demonstrated success in acute depletion of sterols in both plant and animal cells (López et al, 2011; Gerbeau-Pissot et al, 2014; Valitova et al, 2014; Mahammad & Parmryd, 2015; Huang et al, 2019). Here, we found that M$\beta$CD fully

blocked the DSF-induced inhibition of primary roots and root hairs, suggesting DSF's opposite effects compared with M$\beta$CD on sterol perturbation in Arabidopsis. Consistent with this result, culturing seedlings in the presence of both M$\beta$CD and DSF completely abolished the DSF-triggered increase of lipid components, including phytosterols. The total quantity of measured sterol content in M$\beta$CD and M$\beta$CD+DSF treatment were reduced to 92.4 ± 10.3 nmol g$^{-1}$ and 76.0 ± 10.4 nmol g$^{-1}$, respectively, compared with 138.2 ± 24.6 nmol g$^{-1}$ in DMSO treatment and 366.3 ± 121.1 nmol g$^{-1}$ in DSF treatment (Fig 3F). The fact that DSF + M$\beta$CD treatment not only reduce the sterol accumulation induced by DSF (165% increase in sterol content) but also did lower the sterol content (45% reduction) suggest that the effect of M$\beta$CD is dominant over DSF during the long incubation period.

To further examine the role of phytosterols in DSF-induced inhibition of primary root growth, we performed a growth assay on several Arabidopsis mutants of the sterol biosynthesis pathway (Fig 3G) in the presence of DSF. The smt1-1 and fk-X224 mutants were chosen for this experiment as their growth defects are less severe compared with other sterol mutants. Primary root growth of smt1-1 mutant (defective in C-24 methyltransferase) and the fackel (fk) mutant fk-X224 (defective in sterol C-14 reductase), which are two key early steps of sterol biosynthetic pathway upstream of the branch point (He et al, 2003) (Fig S3E), showed significantly reduced sensitivity to DSF, as represented by a lower inhibition rate than their corresponding wild-type ecotypes (Figs 3G and S3F). Together, these findings support our hypothesis that DSF regulates host development through altering host phytosterols content.

### Alteration of sterol composition phenocopies several DSF-induced defects

As we observed that M$\beta$CD and DSF have opposite effects on Arabidopsis lipidomic profiles in the long-term plant growth assay, we next examined whether an acute sterol removal by M$\beta$CD would restore other DSF-induced phenotypes on plant endocytosis and innate immune responses of Arabidopsis. In the experiments testing DSF perturbation of PTI, following M$\beta$CD pretreatment, the number of FLS2-positive endosomes in FLS2–GFP plants pretreated with DSF increased significantly at 60 min after elicitation with flg22, even though M$\beta$CD itself strongly inhibited FLS2 trafficking into endosomes (Figs 4A and S4A). In line with this finding, M$\beta$CD also restored the bulk endocytosis of FM4-64 dye that was blocked by the combined treatment of DSF and ES9-17, represented by the similar ratio of intracellular/PM signal intensity and BFA bodies area between DMSO and ES9-17 treatment in the DSF+M$\beta$CD pretreated seedlings (Figs 4B and S4B), the delay of CLC lifetime (Fig S4C), and the DSF-induced inhibition of ROS burst (Fig 4C).

We further asked whether the supplementation of phytosterols could phenocopy the DSF-induced defects in CME and flg22-activated FLS2 internalization. We found that exogenous phytosterol mix (sitosterol: stigmasterol: campesterol = 8:1:1) treatment phenocopied DSF-induced inhibition of flg22-stimulated ROS production and subsequent M$\beta$CD supplement reversed the sterol-inhibited endocytosis of FM4-64 dye in the presence of ES9-17 (Fig S4D and E), FLS2 internalization into endosomes (Fig S5A), as well as

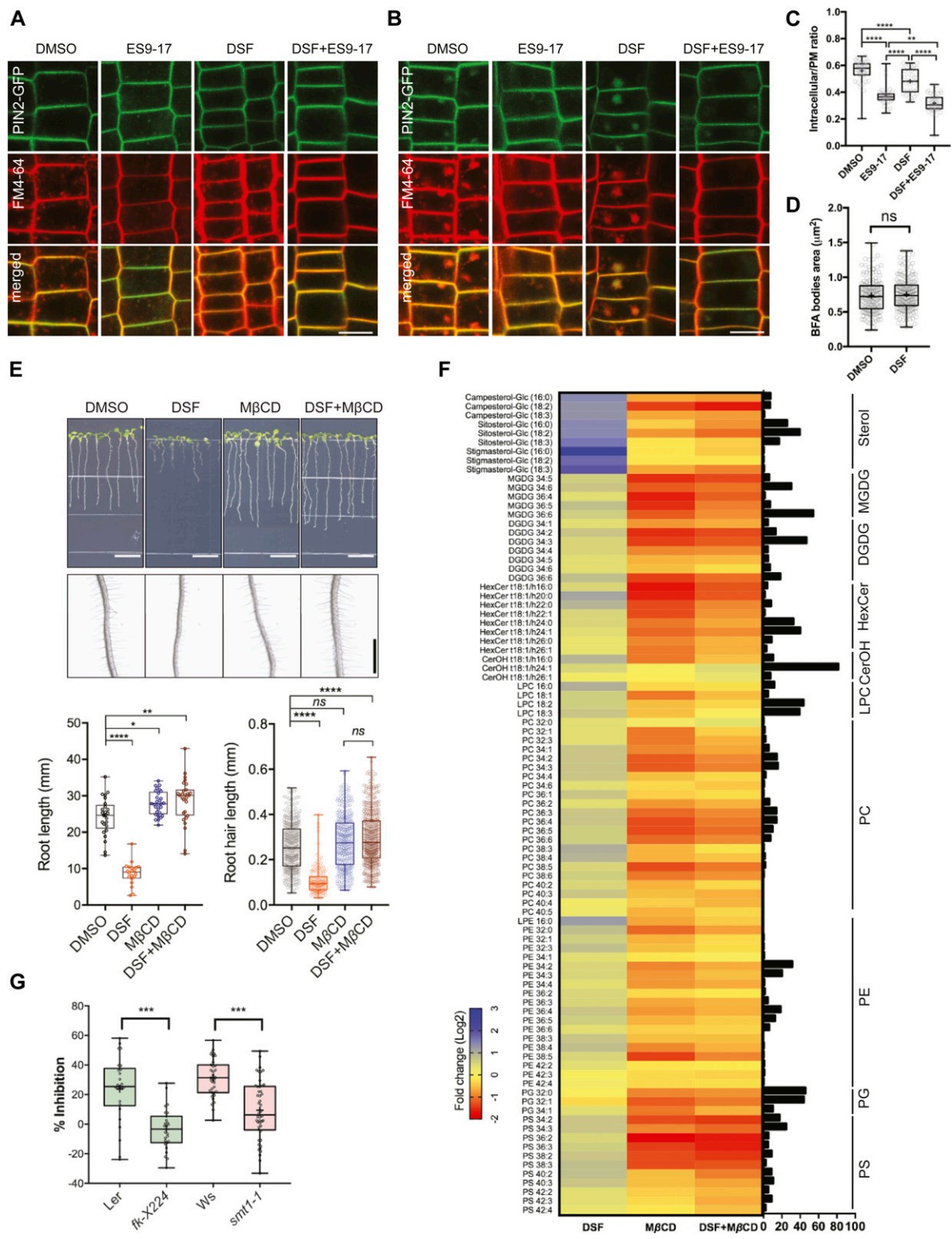

**Figure 3. Diffusible signal factor (DSF) inhibited endocytosis in *Arabidopsis* roots by altering plant lipid profile.**
**(A)** Internalization of the endocytic tracer FM4-64 in PIN2–GFP seedling roots pretreated with DMSO or DSF for 24 h. Plants were treated in liquid 1/2 MS with DMSO (control) or 50 $\mu$M ES9-17 for 1 h before being stained with FM4-64 and visualized using confocal microscopy (n ≥ 5 plants; bars, 10 $\mu$m). **(B)** Accumulation of intracellular FM4-64 signals into BFA compartments after BFA treatment. PIN2–GFP seedlings pretreated for 24 h on 1/2 MS agar supplemented with DMSO or DSF were subjected to BFA or ES9-17+BFA treatments before being stained with FM4-64 and imaged (n ≥ 5 plants; bars, 10 $\mu$m). **(C)** Boxplot represents the ratio of intracellular/plasma membrane signal intensity of FM4-64 dye in root cells presented in Fig 3A as an indicator of bulk endocytosis (n ≥ 37 cells from n ≥ 5 plants). **(D)** Boxplot represents the area of BFA

flg22-triggered ROS burst (Fig S5B). Similar to the sterol depleting compound M$\beta$CD, we observed that Lovastatin—a sterol biosynthesis inhibitor—also rescued ROS burst in DSF-treated leaves (Fig S5C), suggesting that the DSF-induced inhibition of ROS and innate immunity was likely due to the incorporation of DSF into phytosterol metabolic pathways or induction of sterol production via an unknown mechanism. Similar inhibitions of plant root growth were also observed with the predominant sterol $\beta$-sitosterol. $\beta$-Sitosterol could also inhibit root growth in a dose-dependent manner similar to DSF, and this inhibition could also be reverted by M$\beta$CD treatment (Fig S5D and E). Together, these data suggest that a balance in sterol composition of the plant PM is crucial for plant CME as well as the pattern-recognition receptor (PRR)-mediated defense responses.

### DSF and sterol interfere with lateral membrane compartmentalization and FLS2-nano-clustering

Lipid composition, in particular, the composition of sterols in the PM regulates critical physicochemical properties such as lateral motility, protein–protein interaction, and distribution of immune receptors, including FLS2 (Jarsch et al, 2014; Bücherl et al, 2017). Sterols are constituents of the liquid-ordered (L$_o$) lipid nanodomains and influence membrane organization (Grosjean et al, 2015). We, therefore, asked whether the DSF-induced phytosterol increase influences the lateral compartmentalization of the PM by imaging a specific membrane nanodomain marker YFP:REM1.2 (Jarsch et al, 2014). *Arabidopsis* expressing YFP:REM1.2 treated with DSF, a phytosterol mix (sitosterol: stigmasterol: campesterol = 8:1:1) or M$\beta$CD were examined via super-resolution 2D-Structured Illumination Microscopy (2D-SIM). M$\beta$CD resulted in a significant reduction of the mean intensity of YFP:REM1.2-marked nanodomains, suggesting a reduction in the membrane microdomain marked by REM1.2. The presence of either DSF or exogenous sterol could similarly reverse the effect of M$\beta$CD in reducing membrane compartmentalization (Fig 5A and B). Although we only observed the increase of membrane microdomain by exogenous sterol but not DSF itself, it could be a result of the dose-dependency of this particular read-out on microdomain clustering that exogenous sterol exerted on plant membrane compared with DSF treatment. Furthermore, with M$\beta$CD posttreatment, DSF no longer attenuated the FLS2 nano-clusters upon flg22 elicitation, indicated by an anisotropy reduction in living cell homo-FRET imaging which resembles flg22-elicited anisotropy reduction (Fig 5C and D). Together, these data suggest that DSF alters the sterol composition of the PM

resulting in a reorganization of the PM's capacity to build and maintain membrane nanodomains.

## Discussion

Hosts and symbionts constantly exchange small metabolites during interkingdom interactions. For example, the plant pathogenic bacterium *P. syringae* produces the phytotoxin coronatine to target multiple plant immunity pathways, including stomatal defense, thus facilitating bacterial entry as well as disease development (Brooks et al, 2004; Geng et al, 2012; Toum et al, 2016; Song et al, 2019). Conversely, the transfer of small molecules from plants to microbes is also an integral part of microbial symbiotic relationships (Keymer et al, 2017; Cai et al, 2018). During host–pathogen interaction, bacteria mainly rely on QS to regulate the expression of virulence genes necessary for colonization and pathogenesis or to switch between different lifestyles (Gudesblat et al, 2009; He et al, 2010; Khokhani et al, 2017). Numerous reports highlighted the effects of QS molecules on plant development and immunity, but mechanistic insights on how bacterial QS molecules specifically interfere with host biology remain sparse (Hartmann et al, 2015). Distinctive acylation and hydroxylation patterns in bacterial metabolites have been suggested as a molecular signature of AHL QS family to regulate host immune responses (Kutschera et al, 2019; Song et al, 2019). For example, medium-chained 3-hydroxy fatty acids directly bind and trigger the phosphorylation of LIPOOLIGOSACCHARIDE-SPECIFIC REDUCED ELICITATION (LORE) receptor kinase, leading to activation of cytoplasmic receptor-like kinases and stimulate *FRK1* expression (Kutschera et al, 2019; Luo et al, 2020). Here, we unraveled a different mechanism by which a unique QS molecule called DSF regulates host biology by perturbing host phytosterol homeostasis. Unglycosylated sterols and glycosylated sterols (including steryl glycosides and acyl sterol glycosides) are integral components of the plant PMs, especially in membrane nanodomains (Laloi et al, 2007; Ferrer et al, 2017). The increase in host acyl-glycosylated phytosterols by DSF treatment shown in our targeted lipidomic analysis might be a consequence of direct sterol modifications—such as DSF being metabolized to a precursor of acyl source for steryl glucoside acyltransferase (Potocka & Zimowski, 2008), or alteration in potential sterol exchange between PM and ER through PM–ER contact sites (Gatta et al, 2015; Naito et al, 2019)—both of which require further investigation. Regardless of the mechanism, it appears that DSF modulates the sterol composition at the cell surface, and consequently interferes

---

compartments in seedlings grown on DMSO- or DSF-supplemented plates as presented in Fig 3B, n ≥ 161 BFA compartments from n ≥ 5 plants. The FM4-64 uptake experiments were repeated twice with a similar result. **(E)** Growth of Col-0 seedlings on 1/2 MS agar supplemented with DMSO, 25 $\mu$M DSF, 2 mM M$\beta$CD, and 25 $\mu$M DSF+2 mM M$\beta$CD. Images of whole seedlings and root hairs were taken after 7 d (top panel bars, 10 mm; bottom panel bar, 1 mm). The length of primary roots (n ≥ 20 seedlings) and root hairs (n ≥ 140) of 7-d-old seedlings on different media measured by Fiji. The experiment was repeated at least three times with a similar result. **(F)** Lipidomic profiles of *Arabidopsis* Col-0 seedlings after 7 d growing on 1/2 MS agar supplemented with either 25 $\mu$M DSF, 2 mM M$\beta$CD, or 25 $\mu$M DSF + 2 mM M$\beta$CD. Heat map representing the log$_2$ value of fold change compared with DMSO treatment (n = 6 biological replicates). The bar graph parallel to the heat map indicates the relative abundance (in percentages) of the corresponding species within each lipid class. **(G)** DSF-induced inhibition rate of sterol mutants *fk-X224* and *smt1-1* and their corresponding wild-type ecotypes Ler and Ws. Seedlings were monitored for 7 d on DMSO- and 25 $\mu$M DSF-supplemented 1/2 MS plates (n ≥ 28 plants). Inhibition rate (%) was represented by the ratio between the difference in primary root length of seedlings on control (DMSO) versus DSF-supplemented medium and the mean primary root length of control seedlings. Growth assay with sterol mutant was repeated twice with a similar result. *P*-values were determined by one-way ANOVA (*$P < 0.05$; **$P < 0.01$; ***$P < 0.001$; ****$P < 0.0001$; ns, not significant).

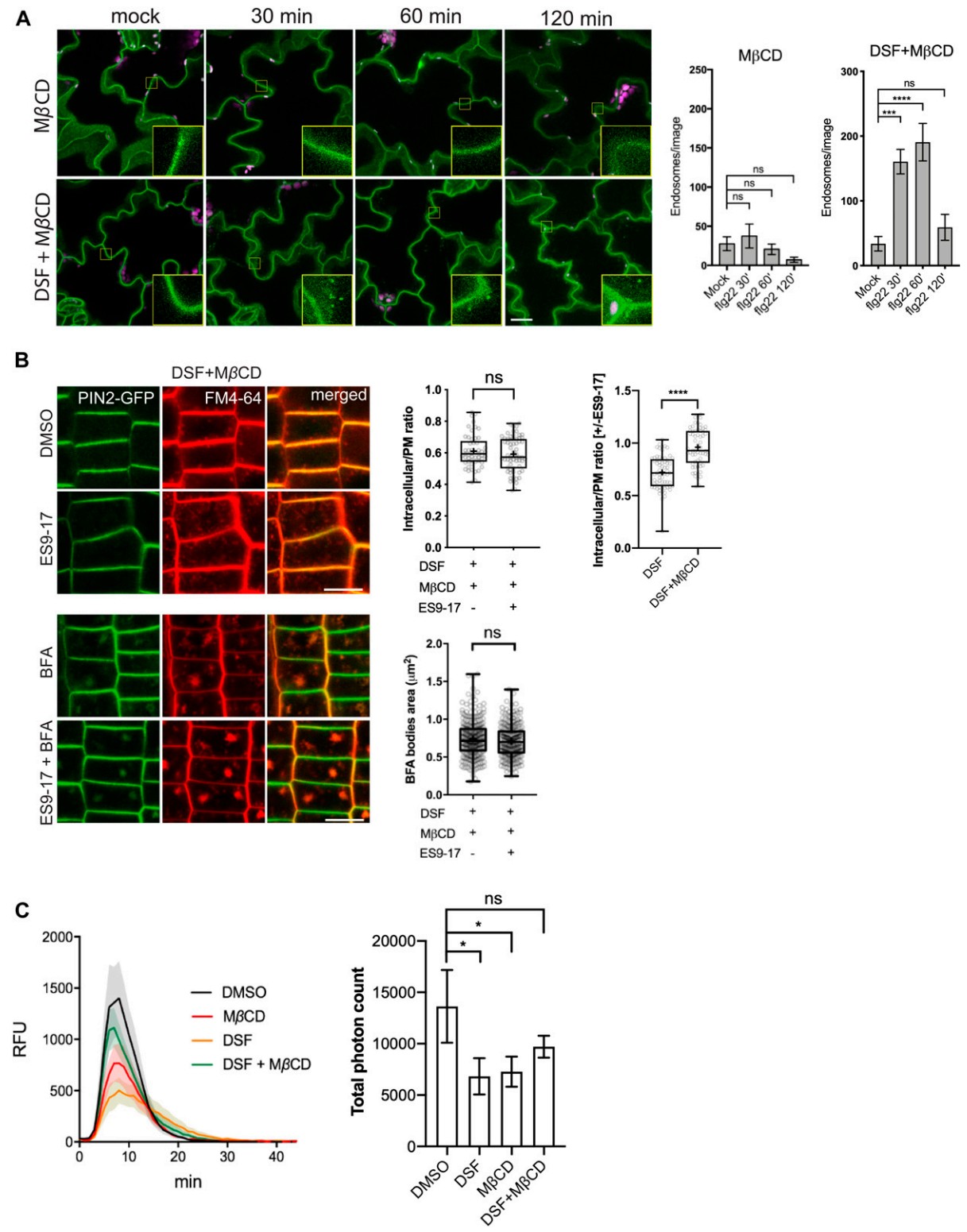

**Figure 4. Sterol depletion by MβCD recovered diffusible signal factor (DSF)/sterol-induced phenotypes on *Arabidopsis* seedlings.**
**(A)** FLS2 receptor internalization into endosomes monitored by confocal microscopy following 10 μM flg22 treatment. FLS2–GFP seedlings were pretreated for 24 h with DMSO or 25 μM DSF in 1/2 MS, followed by a brief 2 mM MβCD treatment for 30 min before peptide elicitation. Micrographs represent maximum projections from 12 slices taken every 1-μm z-distance. Bar graphs showed the number of endosomes per image (n ≥ 10 images from three individual seedlings; bar, 20 μm). The experiment was repeated twice with a similar result. **(B)** Internalization of FM4-64 dye of PIN2–GFP seedling roots after plants were treated for 24 h on 1/2 MS plates with 25 μM DSF followed by 30-min treatment with 2 mM MβCD in liquid 1/2 MS. Plants were treated with 50 μM ES9-17 for 1 h and/or 50 μM BFA for 1 h before staining with FM4-64, and

with plant PTI via the alteration of both the clustering and endocytic internalization of the surface receptors. Compared with untreated plants, DSF-treated plants display an apparent increase in total phytosterol (>2.5-fold), as well as slight enhancement in several other lipid species that structural constituents of the PM, such as glycerolipids (PC, PE, PG, and PS) and sphingolipids (ceramides) (Sperling & Heinz, 2003; Furt et al, 2011; Michaelson et al, 2016). Phosphatidylserine (PS) is known to play an essential function in surface membrane compartmentalization and protein clustering (Raghupathy et al, 2015; Platre et al, 2019). Thus, how precisely the changes of phytosterols and PS by DSF tune the PM mechanical properties will be a subject of future studies using artificial lipid bilayer–based assays and membrane modeling.

The diverse roles of QS molecules from pathogenic bacteria in regulating bacterial responses and modulating host biology reflect a direct co-evolution between hosts and pathogens, according to the Zig-zag model (Jones & Dangl, 2006). *Xcc* DSF gradually accumulates during infection and can reach substantial concentrations (100 μM amount) at later stages of disease (Kakkar et al, 2015), similar to QS molecules from other bacteria (Charlton et al, 2000). A previous report demonstrated that at high concentrations (e.g., 100 μM), DSF acts as a hypersensitive reaction elicitor evident by the cell-death response and excessive callose deposition (Kakkar et al, 2015). Here, we found that at low concentrations (e.g., 25 μM), which is more relevant to the early stages of infection, this QS signal seems to regulate a specific branch of plant PTI responses through the modulation of molecular dynamics and organization of PRRs on the PM. These findings of plant responses to low dose and the high dose of DSF treatments suggests the temporal-regulation of plant host by DSF over the accumulation at different stages of bacteria–host interaction. In addition, the high-dose DSF-induced plant hypersensitive reaction could also be masked by bacterial extracellular polysaccharide during plant colonization, demonstrating the complex bidirectional interaction between plant and bacteria (Kakkar et al, 2015). Taken together, these above results reflect the dynamic change in balancing the tug-of-war over the accumulation of DSF molecules during bacterial infection.

Although we found that DSF did not directly alter PAMP-activated MAPK signaling, heterodimerization of FLS2-BAK1, or *FRK1* expression, this QS molecule did compromise the PAMP-triggered ROS production. Such different DSF-mediated host signaling demonstrates the uncoupled mechanisms of MAPK activation and ROS production upon flg22 elicitation, in which ROS production was attenuated by DSF/sterol, whereas activation of MAPK cascade remained. As a result of a functional MAPK cascade, DSF could only reduce but not entirely abolish the flg22-induced defense responses (Fig 1D and E). FLS2–BAK1 may not be the only rate-limiting components of the defense signaling cascade as *bak1* null mutants still retain a certain level of MAPK and ROS responses (Heese et al, 2007), and FLS2–BAK1 association could be uncoupled from MAPK

activation as previously shown with the flg22 G/A mutant peptide (Sun et al, 2013). Similarly, in chitin-mediated PTI, AtCERK1-mediated MAPK-activation and ROS production were also found to be distinctly regulated by receptor-like cytoplasmic kinases PBL27 and BIK1, respectively (Shinya et al, 2014; Yamada et al, 2016). These observations suggest potential heterogeneity in receptor coupling and distinct constituents in receptor clusters that may convey differential immune response by recruiting different PTI signaling molecules. Similar scenarios have also been observed in the mammalian innate immune system (Dustin & Groves, 2012).

The FLS2 binding partners for MAPK and ROS signaling have differential sensitivity to DSF/sterol-induced perturbation of PM, indicating their possibly different affinity or participation in nanodomains and FLS2 nanoclusters during PRR signaling. For example, because of the active involvement in the high-ordered membrane domain of NADPH oxidase proteins, the NADPH oxidase activity has also been reported to be highly sensitive to plant sterol changes (Lherminier et al, 2009; Liu et al, 2009; Simon-Plas et al, 2011; Hao et al, 2014). Thus, the decline of ROS burst in response to different PAMPs under DSF treatment might also be through the change in signaling competency of other components downstream of FLS2 and BAK1, such as BIK1, BSK1, or heterotrimeric G protein subunits (Zhang et al, 2010; Shi et al, 2013; Hao et al, 2014; Liang et al, 2016; Xu et al, 2017) or the function of the membrane-bound RBOHD itself through several distinct mechanisms (Kadota et al, 2014; Li et al, 2014; Monaghan et al, 2014; Liang et al, 2016; Wang et al, 2018). Of note, the loss-of-function alleles of some of these components have also been documented to affect the flg22-induced ROS burst but not MAPK activation (Zhang et al, 2010; Shi et al, 2013; Liang et al, 2016; Xue et al, 2019). In addition, we observed that MβCD treatment reduced ROS burst at 2 mM, whereas the enhancement of ROS by MβCD was reported in a previous study at 10 mM (Cui et al, 2018). This discrepancy might be due to unknown mechanism in regulating ROS production by MβCD at different doses. The effect of MβCD dosage on plant cell biology, especially the disruption of membrane integrity, membrane compartmentalization, NADPH oxidases behaviors, and ROS production, therefore, needs to be examined empirically in future studies.

Increasing evidence supports the roles of membrane nanodomains in host innate immunity by regulating surface receptor dynamics in and out of nanodomains on PM upon ligand recognition, subsequent receptor interactions, and innate immunity activation (Petrie et al, 2000; Wang et al, 2015; Liang et al, 2018). For instance, disruption of PM continuity is competent enough to abolish the clustering behavior of multiple PM receptors (BRI1, FLS2, and EGFR) (Gao et al, 2015; Bücherl et al, 2017). We found that DSF induced a reduction in SCI by itself and even a further reduction in combination with flg22 (Fig 2B). This could be the result of the additive effect of sterol insertion into PM that perturbed surface membrane compartmentalization and the remaining endocytic

DMSO was used as solvent control (n ≥ 5 plants; bar, 10 μm). Quantification of intracellular/plasma membrane ratio was performed on n ≥ 50 cells from n ≥ 5 plants, and BFA bodies area measurement was performed on n ≥ 250 BFA bodies from n ≥ 5 plants. The experiment was repeated twice with a similar result. The ratio of intracellular/plasma membrane in treatment with ES9-17 versus without ES9-17 treatments of the DSF+MβCD–treated plants was compared with that of the DSF-treated plants presented in Fig 3C. **(C)** Reactive oxygen species production and total photon count of reactive oxygen species burst from Col-0 leaf strips treated for 24 h with DMSO or 25 μM DSF, followed by a 30-min treatment with water (control) or 2 mM MβCD before elicitation by 1 μM flg22 peptide (n ≥ 5/treatment). The experiments were repeated twice with a similar result. *P*-values were determined by two-tailed *t* test (*$P < 0.05$; **$P < 0.01$; ***$P < 0.001$; ****$P < 0.0001$; ns, not significant).

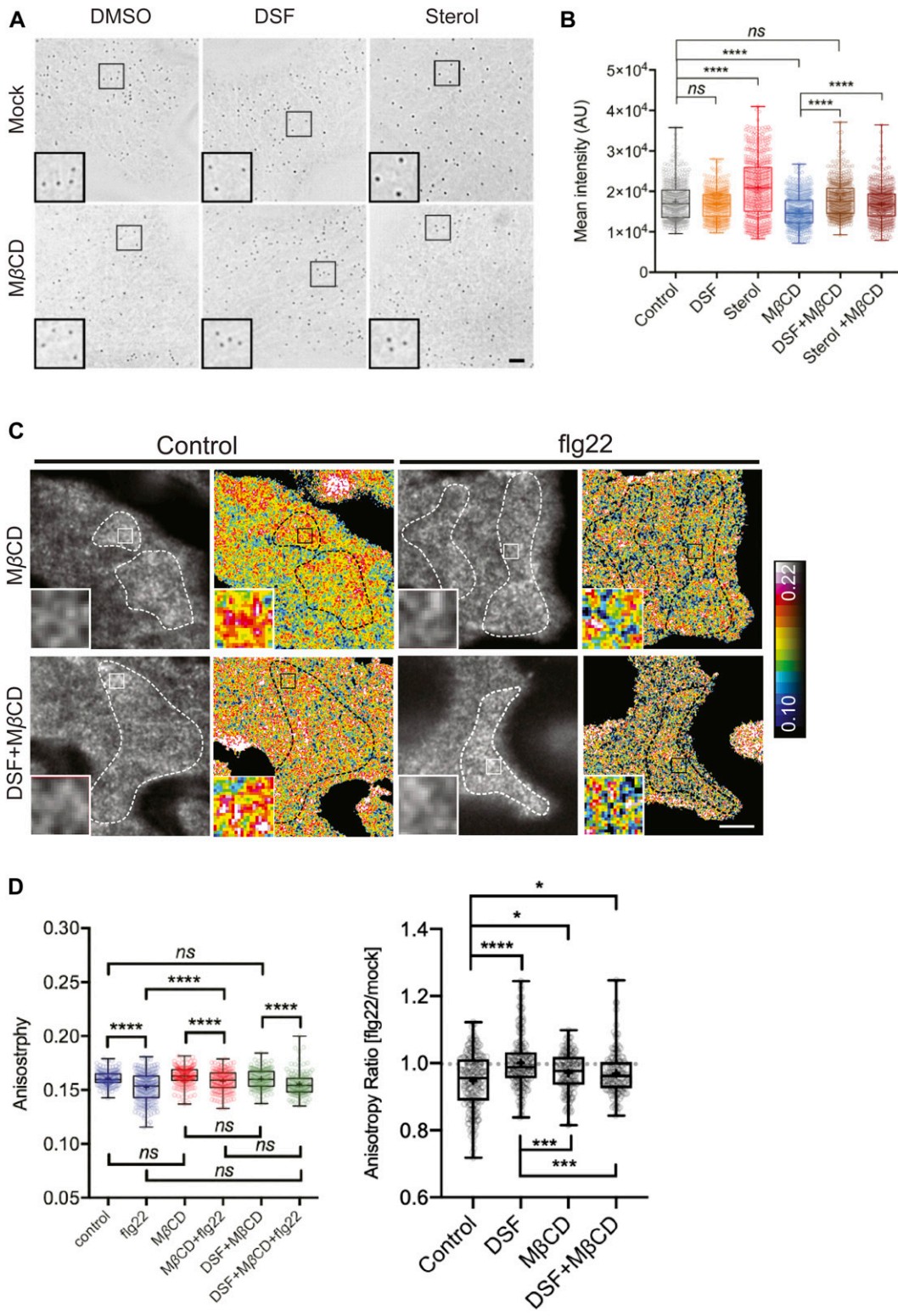

**Figure 5. Lipid nanodomain integrity is dependent on proper sterol composition of the plasma membrane.**
**(A)** 2D-Structured Illumination Microscopy images of the nanodomain marker REM1.2-GFP taken after 24-h treatment with DMSO, diffusible signal factor (DSF), or sterol mix followed by a 30-min treatment with 2 mM MβCD or water control (bar, 2 μm). **(B)** Quantification of Remorin foci intensity on plasma membrane (n = 500 foci from ≥10 cells were analyzed). **(C)** Self-clustering state of FLS2–GFP receptors before and after 5 min of elicitation with 10 μM flg22 following treatment of the seedlings MβCD or DSF+ MβCD. Representative intensity and anisotropy images are shown. Insets are representative 20 × 20-pixel ROIs used for data analysis (bar, 5 μm). **(D)** Boxplot represents anisotropy values measured from Homo-FRET experiment (n ≥ 150 ROIs from eight images acquired from n ≥ 3 seedlings). Anisotropy ratio (flg22/mock) was calculated by dividing the anisotropy value in flg22 treatment to that of water control. The experiment was repeated twice. *P*-values were determined by one-way ANOVA (\**P* < 0.05; \*\**P* < 0.01; \*\*\**P* < 0.001; \*\*\*\**P* < 0.0001; ns, not significant).

activity still facilitated the internalization of FLS2 receptor, both of which retard the FLS2 clustering pattern on the PM. In addition, this drastic reduction in FLS2 clustering could stem from the quantum yield bias with lower absorption signals (less-clustered foci), in which the detectable photon under particular imaging conditions could be shifted by a factor of two or more between highly clustered and diffused foci (van Dam et al, 2019). Our living cell homo-FRET approach allowed us to quantitatively measure the dynamic formation of the nanoclusters of plant surface molecules (Kalappurakkal et al, 2019). Although there are conflicting opinions regarding the association of FLS2 molecules (Ali et al, 2007; Sun et al, 2012, 2013; Somssich et al, 2015), the decrease in anisotropy in homo-FRET suggests the spatiotemporal arrest of FLS2 at PM, which could activate PTI signaling, directly or indirectly, by modulating its interacting partners, thus innate immunity cascades.

Our data suggest a plausible model (Fig 6) in which the *Xcc* QS molecule DSF alters sterol composition and, thereby, modulate membrane nanodomains. This modulation of membrane affects both FLS2 nano-clustering upon PAMP stimulation, general endocytosis pathway, and ROS production. (Raghupathy et al, 2015). Sterols, together with sphingolipids, are major constituents of PM lipid raft/nanodomains that are formed by diverse mechanisms of molecular assemblies of surface molecules (Raghupathy et al, 2015; Köster & Mayor, 2016; Sezgin et al, 2017). Distinct nanodomains spatially separate from each other and their assembly and distributions are of significant importance for the regulation of diverse cellular processes, including the nano-clustering or CME of surface signaling factors (Simons & Ikonen, 1997; Men et al, 2008; Cui et al, 2018). In agreement with this model and our data, sterol content has previously been found to directly or indirectly influence membrane organization and CME (Men et al, 2008; Grosjean et al, 2015; Kim et al, 2017; Cui et al, 2018). CME is well known to coordinate PTI signaling in plants (Ben Khaled et al, 2015; Mbengue et al, 2016). However, as ligand-induced MAPK activation (~5–30 min post elicitation) is detected earlier than the appearance of endocytosis-mediated PRR accumulation in endosomes (~30 min–1 h post elicitation) (Robatzek et al, 2006; Sun et al, 2013) (Fig 2A), endocytic invagination per se might not be a direct mediator for MAPK activation (Smith et al, 2014), which is also supported by the fact here that an attenuated endocytosis by DSF did not alter flg22-mediated MAPK activation, although it reduced ROS production. Nevertheless, the impairment of endocytosis could still influence the internalization and recycling of the surface PRRs at different oligomerization states upon PAMP elicitation (Zou & Ting, 2011; Smith et al, 2014). Consistent with this notion, recent work also highlighted many consequences of the CME impairment in the *Arabidopsis* clathrin mutant *chc2*, such as FLS2 endocytosis, stomatal defense, and callose deposition, increased susceptibility to bacterial infection, whereas acute MAPK response remained intact (Mbengue et al, 2016), as well as the uncoupling of ROS response and MAPK activation (Ranf et al, 2011; Segonzac et al, 2011; Xu et al, 2014).

During bacterial propagation from early- to late stage of infection, DSF attenuates the continuous enhancement of the host PTI responses in ROS production that could otherwise be stimulated by the increasing amount of PAMPs. Receptor activities in both lateral motility for nano-clustering or internalization for receptor recycling on the PM are compromised by DSF treatment. Our report emphasizes the role of QS molecules beyond intraspecies communication, which has important implications for plant disease control (Newman et al, 2004).

## Materials

### Plant growth

*Arabidopsis* (*A. thaliana* [L.] Heynh.) plants were maintained in growth chamber at 22°C under long-day condition (16 h light/8 h dark). For *Arabidopsis* plants used in ROS assay and Western blot, plants were kept at short-day condition (8 h light/16 h dark) to facilitate vegetative growth. Col-0, RabF2b-YFP (Wave 2Y), and Got1p homolog-YFP (Wave 22Y) seeds were obtained from the Arabidopsis Resource Center (ABRC) stock center. *Arabidopsis* FLS2-GFP,

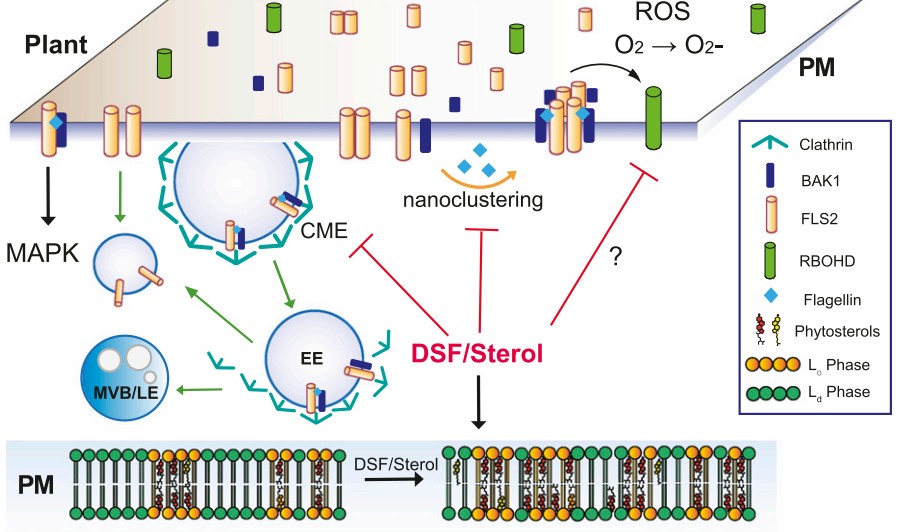

**Figure 6. Proposed model for diffusible signal factor (DSF)–induced suppression of plant innate immunity by perturbation of plasma membrane (PM) lipids.**

Upon contact with bacteria/flagellin, FLS2 receptors form heterodimers with interacting partners on the PM and also increased its nano-clustering. Activated receptors then relay the signals to PM co-receptors and cytoplasmic kinases to induce a wide range of defense responses and also undergo endocytosis. In plants infected with DSF-producing bacteria, DSF could be metabolized into plant sterols or directly induce the production of phytosterols via unknown mechanism(s). Sterol enrichment on the plant PM leads to an increase in lipid ordered phase—$L_o$ (or PM nanodomains) and a decrease in lipid disordered phase ($L_d$ phase), and receptor nano-clustering was affected as a result. The behaviors of other pattern-recognition receptors and reactive oxygen species production mediated by RBOHD activity could potentially be impaired by DSF via the disruption of the protein clustering on plant PM. Therefore, signaling cascades from PM that result in PTI responses were attenuated.

pREM1.2::YFP:REM1.2, CLC2-GFP, BRI1-GFP, and BOR1-GFP marker lines were described previously (Robatzek et al, 2006; Geldner et al, 2007; Fan et al, 2013; Jarsch et al, 2014; Shen et al, 2018).

Sterol mutants *fkX224*, *smt1-1*, and their WT ecotypes were described previously (Diener et al, 2000; Schrick et al, 2000). For cell biology imaging, *Arabidopsis* seeds were surface sterilized with 10% bleach and 70% EtOH, washed three times with sterilized water, and vernalized at 4°C for 2 d. Seedlings were then sown on 1/2 MS agar and grown at 22°C for 4–5 d under long-day condition (16 h light/8 h dark) before treatments and imaging. For growth assay to evaluate primary root growth, we germinated surface-sterilized seeds directly on 1/2 MS agar containing indicated drugs. Plants were grown vertically at 22°C for 7 d before images of the plates were taken using a flatbed scanner, and primary root length was determined by Fiji software. Unless specified, all growth assays were repeated at least twice with ~20–30 seedlings for each treatment. For growth assay of sterol biosynthesis mutants on DMSO- and DSF-supplemented medium, growth inhibition rate (%) were defined as the percent difference between the root length of DSF-treated plants and the mean root length of DMSO-treated plants.

### Bacterial strains

*P. syringae* pv. *tomato* DC3000 (*Pst*) (Cuppels, 1986) were maintained regularly at 28°C on nutrient yeast glycerol plates.

# Methods

### Chemicals and treatment conditions

DSF (*cis*-11-methyl-2-dodecenonic) was purchased from Merck and dissolved in DMSO to obtain 100 mM stock. The bacterial flagellin peptides: flg22 and flgII-28, as well as elf26 peptide were chemically synthesized (GL Biochem Ltd.) with 95% purity and dissolved in water to obtain 5 mM stocks. For pharmacological studies, Brefeldin A (BFA), Wortmannin, Lovastatin, and ES9-17 were dissolved in DMSO as 50 mM stocks. Unless specified, seedlings were incubated with these drugs at the working concentration of 50 $\mu$M in 1/2 MS medium for 1 h at room temperature. The sterol-depleting reagent M$\beta$CD was dissolved in deionized water at 200 mM and filter-sterilized. For short-term M$\beta$CD treatment, *Arabidopsis* seedlings were treated with 2 mM M$\beta$CD in liquid 1/2 MS for 30 min at room temperature. Long-term growth assay with M$\beta$CD was performed by germinating axenic *Arabidopsis* seeds directly on 1/2 MS agar containing 2 mM M$\beta$CD.

### qRT-PCR

*Arabidopsis* Col-0 seeds were surface-sterilized and vernalized as mentioned above. 5-d old seedlings were transferred to new 1/2 MS plates containing either DMSO or 25 $\mu$M DSF for three additional days. At day 8, the seedlings were subjected to a brief treatment with either 150 nM of flg22 or water control, and plant tissue was frozen in liquid nitrogen and kept at −80°C. RNA extraction was performed using RNeasy Plant Mini Kit (QIAGEN) according to the

manufacturer's instruction. Each treatment includes 2–3 biological replicates.

DNase treatment was performed twice using Turbo DNase (Ambion) and RNAClean XP kit (Agencourt). The quality of RNA samples was evaluated using the Qubit Fluorometer (Thermo Fisher Scientific). cDNA was synthesized by SuperScript III First-Strand Synthesis system (Invitrogen), using oligo-d(T) primer. Quantitative RT-PCR was performed in triplicate for each of the biological replicates, using Kapa SYBR FAST qPCR Master Mix with primers specific for *FRK1* (He et al, 2006), and *EF-1α* gene used as an internal control (Czechowski et al, 2005). Data were collected by Applied Biosystem StepOnePlus Real-Time PCR system (Applied Biosystem) and analyzed using the manufacturer's software.

### Microscopy and data analysis

#### *Confocal microscopy*

Confocal microscopy was performed on a Zeiss ELYRA PS.1 + LSM780 system equipped with a 63× Plan-Apochromat oil-objective (NA = 1.4). GFP/YFP and FM4-64 were excited with a 488- and 514-nm laser, respectively. Emission was collected at 493–594 nm for GFP/YFP and 612–758 nm for FM4-64. In addition, chloroplast autofluorescence in FLS2-GFP cotyledons was also captured from 630 to 730 nm.

For imaging of FLS2-containing endosomes, image acquisition was performed as described previously (Leslie & Heese, 2017). Drug treatments of FLS2–GFP seedlings were performed in liquid 1/2 MS as follows: DSF: 25 $\mu$M, 24 h; sterol mix (sitosterol: stigmasterol: campesterol = 8:1:1): 50 $\mu$M, 24 h; M$\beta$CD: 2 mM, 30 min (alone or in combination with 25 $\mu$M DSF-24 h); and 10 mM, 24 h (alone or in combination with 50 $\mu$M sterol mix, 24 h). After drug treatments, FLS2–GFP seedlings were further elicited with 10 $\mu$M flg22 and imaged at 30′, 60′, and 120′ post-elicitation (water was used as mock control). We also included at treatment of 10 $\mu$M flgII-28 as negative control because flgII-28 peptide is not recognized by the *Arabidopsis* FLS2 receptor. At indicated time points, z-stack images of FLS2–GFP seedling cotyledons were taken at 12 slices of 1-$\mu$m z-distance. Endosome signals from maximum projection images were segmented to remove PM signal using Fiji "Trainable Weka Segmentation" plug-in. Segmented images were analyzed further by "Particle Analysis" tool in Fiji (Schindelin et al, 2012) to obtain the number of endosomes per image. Experiments were repeated twice, with 12 images taken from three seedlings for each replicate.

To monitor FM4-64 uptakes in PIN2-GFP roots, seeds were sown on 1/2 MS agar and grown vertically. 5-d-old PIN2–GFP seedlings were then transferred to a new plate containing either DMSO (control), 25 $\mu$M DSF, or 50 $\mu$M sterol mix (sitosterol: stigmasterol: campesterol = 8:1:1) for an additional day. After 24 h of growth on DMSO/DSF/sterol–supplemented medium, the following drug treatments were performed in liquid 1/2 MS at room temperature: M$\beta$CD: 2 mM, 30 min (alone or in combination with 25 $\mu$M DSF, 24-h treatment); M$\beta$CD: 10 mM, 24 h (alone or in combination with 50 $\mu$M sterol mix, 24-h treatment); ES9-17: 50 $\mu$M, 1 h; and BFA: 50 $\mu$M, 1 h. After treatments, PIN2–GFP seedlings were incubated with 2 $\mu$M FM4-64 for 5 min at room temperature, washed twice with fresh 1/2 MS medium, and roots were imaged immediately using an ELYRA PS.1 + LSM780 system equipped with a 63× Plan-Apochromat oil-objective (NA = 1.4).

### VA-TIRFM and lifetime analysis

Variable-angled total internal reflection fluorescence microscopy (VA-TIRFM) was performed on a Zeiss ELYRA PS.1 + LSM780 system equipped with a 100× Plan-Apochromat oil-objective (NA = 1.46) to monitor the dynamics of PM markers. Time-lapse images of FLS2–GFP foci on PM were taken at up to 120 frames with 1,000-ms exposure at 2% laser power. Time-lapse movies of other PM marker lines were acquired as followed: CLC–GFP: 90 cycles, 1,000-ms exposure, 1,000-ms interval; BRI1–GFP and BOR1–GFP: 90 cycles, 500-ms exposure, 500-ms interval. Acquired VA-TIRFM images were deconvolved using Huygens Essential (Scientific Volume Imaging) using theoretical point spread functions. Particle lifetime was measured from kymographs in Fiji. Lifetime of at least 100 individual foci from three different seedlings was presented. The clustering of FLS2–GFP foci on plant PM, indicated by SCI was measured as described previously (Gronnier et al, 2017). Briefly, SCI was calculated based on the ratio of top 5% pixels with the highest intensity and 5% of pixels with the lowest intensity along 10-$\mu$M line ROIs selected from the first frame of each VA-TIRFM images. Data presented in boxplots were from at least 60 individual ROIs.

### 2D-structure illumination microscopy

2D-SIM image for pREM1.2::YFP:REM1.2 signals was conducted using a Zeiss ELYRA PS.1 + LSM780 system with a Zeiss Alpha Plan Apochromat 100× (NA = 1.46) oil objective. A 488-nm laser light source was used to excite the YFP signal, and the emission was collected in the range of 495–575 nm by a sCMOS camera (PCO) with a pixel size as 6.5 × 6.5 $\mu$m, the exposure time was set as 300 ms. All the 2D-SIM images contained five rotations and five phases of the grated pattern (42 $\mu$m) for each image layer. SIM image reconstruction was conducted by setting the super-resolution frequency weighting as 0.5 and the noise filter as –4.

Particle tracking was performed in Imaris (Bitplane). Imaris Spots function was used to automatically quantify the REM1.2 particle signals. To trace all the spots, the estimated spot diameter was set to 0.22 $\mu$m according to the preliminary particle analysis before processing the whole image, and the threshold was set to automatic region threshold. After tracing all the particles in each image, the total intensity of each particle was analyzed and output automatically. For each treatment, data of 500 particles were randomly selected from more than 10 images (more than 2,000 particles in total for each treatment) for further statistical analysis.

### FLS2–GFP steady-state anisotropy measurement

Steady-state anisotropy was measured on FLS2–GFP–expressed live cells in cotyledon using NikonTE2000 TIRF microscope with dual camera as described previously (Ghosh et al, 2012). Briefly, with respect to the plane of polarization of 488 nm excitation light (~5.2 mW power), emission intensity was collected by two Evolve 512 EMCCD camera at parallel ($I_{pa}$) and perpendicular ($I_{pe}$) direction. PM of the plant was illuminated at HILO using variable TIRF angle module until the best signal-to-noise ratio was achieved using a 100× (NA = 1.45) objective with an image plane pixel size of 106 nm and 1 s exposure time. Anisotropy measured at HILO (1,910 and 1,950 angles) showed the same results as definitive TIRF, using GFP solution and stable transfected GFP–GPI construct in CHO cells (Fig S2D and E). Anisotropy values measured at HILO (2,000 angle) even

showed similar results if ROI were taken from evenly illuminated area of the cells. However, we have restricted our all measurements at HILO angle <1,950 to nullify any bias due to uneven illumination. G-factor of the instrument was calculated with freely diffused FITC solution by calculating ratio of $I_{pa}$ versus $I_{pe}$ emission intensities (Ghosh et al, 2012). Then anisotropy of the FLS2–GFP was calculated by G-factor–corrected perpendicular images using the following equation:

$$r(\text{anisotropy}) = \frac{I_{pa - I_{pe}}}{I_{pa + 2I_{pe}}}$$

Alignment of images of $I_{pa}$ and $I_{pe}$ and G-factor corrections was carried out using algorithm written in MATLAB (MathWorks). FLS2–GFP anisotropy in different treatment conditions was compared using data from more than 150 ROIs (20 × 20 pixel) from at least eight images taken from at least three seedlings. The experiment involved two biological replicates.

### Transmission electron microscopy

*Arabidopsis* Col-0 seeds were surface-sterilized and vernalized at 4°C for 2 d and then grown on 1/2 MS plates at 22°C (16 h light/8 h dark). After 5 d, the seedlings were transferred to a new 1/2 MS plate containing either 25 $\mu$M DSF or DMSO (solvent control) for an additional 3 d.

8-d-old seedlings were severed into root, hypocotyl, and cotyledon sections. Plant tissues were fixed in a primary fixative (1 ml of 2.5% glutaraldehyde prepared in 25 mM sodium cacodylate buffer, pH 7.2) overnight at 4°C. Plant tissues were then washed four times in 25 mM sodium cacodylate buffer, 10 min each time. Then, the plant tissues were submerged in 1 ml of secondary fixative (2% osmium tetraoxide and 0.5% potassium ferricyanide in water) for 2 h at room temperature.

The samples were washed four times with 1 ml of deionized water, each time for 10 min, then transferred to 2% uranyl acetate solution, and incubated at room temperature for 4 h. Plant samples were washed four times with deionized water, each time for 10 min until the solution became clear. Sample dehydration was carried out at room temperature with a series of increasing acetone solution as follows: 30%, 20 min; 50%, 20 min; 75%, 20 min; and 100%, 20 min (2×).

Tissue embedding was performed using Spurr Low-Viscosity Embedding Kit (Sigma-Aldrich). The samples were incubated in increasing Spurr solutions, each time for 1 h as follows: 25%, 50%, 75%, and 100%. After incubating with 100% Spurr, the samples were kept in fresh 100% Spurr solution overnight at 4°C. The next day, fresh 100% Spurr was added to the plant samples one more time. Plant tissues were then transferred into BEEM embedding capsules (Electron Microscopy Sciences) with the position of the samples at the bottom of the capsules, then incubated for 4 h at RT, followed by another 2 d at 60°C. Sectioning was performed using a Cryo Leica EM UC7 diatome (Leica) with the sample thickness set to 70 nm.

Post-staining ultrathin sections were performed by incubating the sections with uranyl acetate (in methanol) for 1 min, and then washed with double-deionized water. The sections were then incubated with Reynold's lead citrate solution for 1 min and

subsequently washed with deionized water, and then dried in a fume hood. The plant tissue sections were imaged using a Hitachi Transmission Electron Microscope HT7700 (Hitachi) with the acceleration voltage of 80 kV.

## Lipidomic profiling

### Lipid extraction

Surface-sterilized Col-0 seeds were directly germinated and grown vertically on 1/2 MS agar supplemented with DMSO, 25 $\mu$M DSF, 2 mM M$\beta$CD, or DSF+M$\beta$CD for 7 d at 22°C, 16 h light/8 h dark. After 1 wk, 30 seedlings were pooled together as one biological replicate and total lipid was extracted as previously described (Vu et al, 2014). Before lipid extraction, 500 pmol each of PC 28:0, PE 28:0, PS 28:0, PG 28:0, PA 28:0, LPC 20:0, LPE 14:0, LPA 17:0, SM 12:0, Cer 17:0, CerOH 12:0, HexCer 8:0, TG 48:0 d5, GB3 23:0, and cholesterol d6 were added to each sample as internal standards. The experiment involved six biological replicates for each treatment.

### LC–MS analysis

Dried lipid extracts were resuspended in 500 $\mu$l butanol–methanol (1+1, vol/vol) and stored at −20°C until LC–MS analysis. Phospholipids, sphingolipids, and sterol lipids were identified using an Agilent 1290 UHPLC system coupled to an Agilent 6495 triple quadrupole mass spectrometer. Chromatographic separation was performed on an Agilent Eclipse Plus C18 reversed phase column (50 × 2.1 mm, particle size 1.8 $\mu$m) using gradient elution of solvent A (water/acetonitrile 60/40, vol/vol, 10 mM ammonium formate) and solvent B (isopropanol/acetonitrile 90/10, vol/vol, 10 mM ammonium formate). 20% solvent B was increased to 60% B over 2 min and to 100% B over 5 min. The column was flushed with 100% B for 2 min, and then re-equilibrated with 20% B for a further 1.8 min. The flow rate was 0.4 ml/min, column temperature was 50°C, and injection volume was 1 $\mu$l. The JetStream ion source was operated at the following conditions: gas temperature, 200°C; gas flow, 12 liters/min; nebulizer gas pressure, 25 $\psi$; sheath gas heater temperature, 400°C; sheath gas flow, 12 arbitrary units; capillary voltage, 3,500 V for positive ion mode and 3,000 V for negative ion mode; Vcharging, 500 for positive ion mode and 1,500 for negative ion mode; positive high pressure RF, 200; negative high pressure RF, 90; positive low pressure RF, 100; negative low pressure RF, 60; fragmentor voltage, 380 V; cell accelerator voltage, 5 V; and dwell time, 1 ms.

The injection order of 24 samples (six replicates of four treatment groups) was randomized and bracketed by injections of pooled quality control and extracted blank samples to monitor system stability and carryover. The whole sample set was analyzed twice consecutively with different selected reaction monitoring (SRM) settings. One injection was for phospholipids and sterols using SRM transitions from Vu et al (2014), and the second method detected plant sphingolipids with SRM transitions adapted from Markham and Jaworski (2007).

### Data processing

Lipids were analyzed based on characteristic SRM transitions and retention time using Agilent MassHunter Quantitative Analysis for QQQ (v. B.07.00). Further processing, including normalization to

suitable internal standards, was performed in Microsoft Excel. High-resolution LC–MS data were processed using XCalibur QuanBrowser 3.0.63 (Thermo Fisher Scientific).

## Immunoblot analysis

Three leaves discs (0.7 cm in diameter) from 5- to 6-wk-old *Arabidopsis* plants were cut into strips and floated for 24 h in sterile water in 24-well plates under constant light to reduce stress due to wounding. DMSO or DSF were added into the wells 24 h before elicitation with 10 $\mu$M flg22. Leaf tissue was washed with sterile water to remove the elicitor and flash-frozen in liquid nitrogen at different time points. Total protein extraction was performed as described previously (Heese et al, 2007).

To detect the flagellin receptor, we used $\alpha$-GFP antibody (1:200; Torrey Pines BioLabs) to probe for the FLS2–GFP in FLS2–GFP transgenic plants. For monitoring p-MAPK response, we used $\alpha$-p-MAPK antibody (1:3,000; Cell Signalling Technology) to detect phosphorylated MAPKs in Col-0 leaf tissue after flg22 peptide elicitation. Western blot membranes were stained with Ponceau S as internal loading control.

## Measurement of callose deposition

Measurement of plant callose deposition in response to bacterial flagellin peptide was performed on 2-wk-old Col-0 seedlings germinated on 1/2 MS medium. Whole seedlings were individually placed in six-well plates with 4 ml of liquid 1/2 MS supplemented with DMSO or 25 $\mu$M DSF for 24 h at 22°C, and then elicited for 24 h with 1 $\mu$M of flg22. Challenged plants were destained with acetic acid:ethanol (1:3) overnight, and then washed in 150 mM $K_2HPO_4$ for 30 min. Destained leaves were incubated in 0.01% aniline blue (dissolved in 150 mM $K_2HPO_4$) in the dark for 2 h, and then mounted on microscope slides in 50% glycerol. Callose deposition were observed on a Zeiss ELYRA PS.1 + LSM780 system using a DAPI filter (excitation:emission, 370:509 nm) with a 10× objective. The experiment was repeated twice, each involved eight individual plants and at least six images were taken from each plant. For callose deposition response to DSF alone, the plants were incubated with increasing concentrations of DSF for 24 h in 1/2 MS, and then stained with aniline blue and visualized as indicated above. The experiment was performed once with n ≥ 18 images taken from three individual plants. Callose deposition was quantified using Fiji software.

## Stomatal closure assay

5-wk-old Col-0 plants were used to determine the stomatal closure response in response to the flg22 peptide. Briefly, two-three leaves/plant were marked with a permanent marker, and then infiltrated with DMSO or 25 $\mu$M DSF (diluted in sterilized 10 mM $MgCl_2$). The plants were returned to the growth chamber for an additional 24 h. The next day, infiltrated leaves were excised from the plants, and the abaxial epidermis was peeled off and transferred to a petri disc containing sterile water using a natural-hair brush (size 8). 10 $\mu$M flg22 was added to the samples for 30 min. The epidermal peels were stained with 20 $\mu$M propidium iodide for 5 min and rinsed

briefly with sterile water, then mounted on microscope slides without coverslips and examined under a wide-field fluorescence microscope (Leica) equipped with RFP filter using a 10× objective lens. The experiments involved measurements of ≥130 stomata from at least 10 images taken from five individual plants.

## ROS production

Measurement of ROS burst in *Arabidopsis* apoplast elicited by flg22 was performed as described previously (Smith et al, 2014). Briefly, each leaf disc (0.7 cm in diameter) from 5- to 6-wk-old *Arabidopsis* plants was cut into five strips, floated on sterile water for 24 h in white 96-well plates under constant light at room temperature to reduce stress from wounding. Drugs were added into the wells before elicitation with 1 $\mu$M flg22 as follows: 25 $\mu$M DSF/DMSO control, 24 h; 2 mM M$\beta$CD, 30 min; 50 $\mu$M sterol mix, 24 h; and 1 $\mu$M lovastatin, 24 h. Each ROS experiment was performed in duplicate, and experiments shown on the same graph were performed in the same plate.

## Plant infection assay

2-wk-old *Arabidopsis* plants grown on 1/2 MS agar was used for flood inoculation with *P. syringae* pv. *tomato* DC3000 (*Pst* DC3000). Using *Pst* DC3000, a bacterial pathogen with a different QS system other than DSF allowed us to dissect the effect of DSF on plant immunity independent of effector proteins activation and other bacterial traits regulated by QS. Briefly, plants were flooded with 40 ml of 10 mM MgCl$_2$ solution containing 25 $\mu$M DSF or DMSO (control). After 1 d, the liquid was decanted and another 40 ml of MgCl$_2$ containing either 1 $\mu$M of flg22 or water was poured into each plate. On the next day, the seedlings were inoculated with 5 × 10$^6$ CFU/ml of *Pst* DC3000 prepared in 10 mM MgCl$_2$. Bacterial population was determined 4 d postinoculation by dilution plating as described previously (Ishiga et al, 2011). The experiment was repeated twice with 9–12 technical replicates/treatment. The relative ratio of log$_{10}$ (CFU/mg) was calculated by dividing the colony count of flg22-treated group to the mean of the control group (water) for each of the drug treatment (DMSO or DSF).

## Co-immunoprecipitation

The constructs for transient expression of BAK1 and FLS2 have been described earlier (Xu et al, 2017). Protoplast-based co-IP assays were conducted as previously described (Cheng et al, 2015). Briefly, 1 ml of transfected protoplasts was incubated for 12 h for protein expression in the presence of 1 $\mu$M DSF (Sigma-Aldrich) and was treated with or without 1 $\mu$M flg22 (MoonBiochem) for 10 min before harvest. Briefly, the cells were lysed in 500 $\mu$l IP buffer (10 mM Hepes, pH 7.5, 100 mM NaCl, 1 mM EDTA, 10% glycerol, 1% Triton X-100, and 1× protease inhibitor cocktail [Roche]), and 30 $\mu$l of protein lysate was kept as the input fraction. The remaining 470 $\mu$l lysate was incubated with 10 $\mu$l anti-FLAG M2 affinity gel (Sigma-Aldrich) at 4°C for 3 h. After being washed five times with ice-cold IP buffer and once with 50 mM Tris–HCl (pH 7.5), the bound proteins were solubilized by boiling the beads in 60 $\mu$l 2× SDS–PAGE loading buffer and were detected by immunoblotting using anti-FLAG

(Sigma-Aldrich) or anti-HA antibody (Roche). The experiment was repeated twice.

## Quantification and statistical analysis

Statistical analyses and data visualization were performed using GraphPad Prism 7.0 (GraphPad). Mean comparisons were performed using one-way ANOVA (*$P < 0.05$; **$P < 0.01$; ***$P < 0.001$; ****$P < 0.0001$; ns, not significant). Unless specified, data were presented as box-and-whisker plots with bars indicating the median, 25% and 75% quartiles.

## Contact for reagent and resource sharing

Material used in this study can be found in the Key Resource Table (Table S1). Further information and requests for reagents and resources should be directed to and will be fulfilled by the Lead Contact, Yansong Miao (yansongm@ntu.edu.sg).

# Supplementary Information

# Acknowledgements

We are grateful to Kimberly Kline (Singapore Centre for Environmental Life Sciences Engineering, Singapore) for critical reading of the manuscript, Yuki Nakamura (Institute of Plant and Microbial Biology, Academia Sinica, Taiwan) for valuable discussion on the lipidomics data. We thank Lay Yin Tang and Alma Turšić-Wunder for helping in RNA preparation. We thank the following researchers for sharing the *Arabidopsis* seeds: Takashi Ueda (National Institute for Basic Biology, Japan) for the FLS2–GFP *Arabidopsis*, Jianwei Pan (Lanzhou University, China) for PIN2–GFP *Arabidopsis*, Liwen Jiang (Chinese University of Hongkong) for the VHAa1–GFP line, and Thomas Ott (University of Freiburg) for the pREM1.2::YFP:REM1.2 line; Kathrin Schrick (Kansas State University) and Jyan-Chyun Jang (Ohio State University) for the *Arabidopsis* sterol mutants; and Jinbo Shen (State Key Laboratory of Subtropical Silviculture, Zhejiang A&F University, China) for the BOR1–GFP line. We also thank Yinyue Deng (South China Agricultural University, China) for sharing reagents; MK Mathew and Divya Rajagopal (National Centre for Biological Sciences-TIFR, India) for helping with *Arabidopsis* growth for the Homo-FRET experiments. S Mayor is supported by a Margdarshi Fellowship (IA/M/15/1/502018) from the Department of Biotechnology-Wellcome Trust Alliance. This study was supported by Nanyang Technological University NIMBELS grant (NIM/01/2016) and Nanyang Technological University startup grant (M4081533) to Y Miao in Singapore.

## Author Contributions

TM Tran: conceptualization, data curation, formal analysis, investigation, visualization, methodology, and writing—original draft, review, and editing.

Z Ma: conceptualization, data curation, formal analysis, visualization, and writing—review and editing.

A Triebl: data curation, formal analysis, investigation, methodology, and writing—review and editing.

S Nath: data curation, formal analysis, investigation, visualization, methodology, and writing—review and editing.

Y Cheng: data curation, formal analysis, and investigation.

B-Q Gong: data curation, formal analysis, and investigation.

X Han: data curation, formal analysis, and investigation.

J Wang: data curation, formal analysis, and investigation.

J-F Li: data curation, formal analysis, investigation, and writing—review and editing.

MR Wenk: data curation, formal analysis, and investigation.

F Torta: conceptualization, data curation, formal analysis, investigation, and writing—review and editing.

S Mayor: conceptualization, data curation, formal analysis, supervision, investigation, and writing—review and editing.

L Yang: conceptualization, data curation, formal analysis, funding acquisition, investigation, and writing—review and editing.

Y Miao: conceptualization, data curation, formal analysis, supervision, funding acquisition, investigation, visualization, methodology, project administration, and writing—original draft, review, and editing.

## Conflict of Interest Statement

The authors declare that they have no conflict of interest.

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
