## [Reviewer comments · Life Science Alliance]

Life Science Alliance

The bacterial QS signal DSF hijacks Arabidopsis sterol biosynthesis to suppress plant immunity

Tuan Tran, Zhiming Ma, Alexander Triebel, Sangeeta Nath, Yingying Cheng, Ben-Qiang Gong, Xiao Han, Junqi Wang, Jian-Feng Li, Markus Wenk, Federico Torta, Satyajit Mayor, Liang Yang, and Yansong Miao

DOI: <https://doi.org/10.26508/lsa.202000720>

Corresponding author(s): Yansong Miao, School of Biological Sciences, Nanyang Technological University, Singapore

Review Timeline:

Submission Date:	2020-03-29
Editorial Decision:	2020-05-18
Revision Received:	2020-07-03
Editorial Decision:	2020-07-22
Revision Received:	2020-07-26
Accepted:	2020-07-30

Transaction Report:

May 18, 2020

Re: Life Science Alliance manuscript #LSA-2020-00720

Prof. Yansong Miao
School of Biological Sciences
Nanyang Technological University
Singapore

Dear Dr. Miao,

Thank you for submitting your manuscript entitled "The bacterial QS signal DSF hijacks Arabidopsis sterol biosynthesis to suppress plant immunity" to Life Science Alliance. The manuscript was assessed by expert reviewers, whose comments are appended to this letter.

As you will see, the reviewers appreciate the aim of your study. However, they also think that more support for your conclusions is needed and that pleiotropic effects may underlie the observed effects. We would like to invite you to submit a revised version of your manuscript to us, addressing the individual concerns raised by the reviewers. Some of the concerns can get addressed by text changes, scholarly citing the existing literature to tie your results better into the existing knowledge, and toning down conclusions. A significant amount of experimental revision work is needed, too, and we concluded that performing the experiments is in principle feasible and necessary.

In our view these revisions should typically be achievable in a few months. However, we are aware that many laboratories cannot function fully during the current COVID-19/SARS-CoV-2 pandemic and therefore encourage you to take the time necessary to revise the manuscript to the extent requested above. We will extend our 'scooping protection policy' to the full revision period required. If you do see another paper with related content published elsewhere, nonetheless contact me immediately so that we can discuss the best way to proceed.

When submitting the revision, please include a letter addressing the reviewers' comments point by

point.

Thank you for this interesting contribution to Life Science Alliance. We are looking forward to receiving your revised manuscript.

Sincerely,

B. MANUSCRIPT ORGANIZATION AND FORMATTING:

Reviewer #1 (Comments to the Authors (Required)):

In their manuscript, Tran and co-authors study the effect of a quorum sensing molecule called diffusible-signal factor (DSF) on plant immune responses triggered by the perception of the epitope flg22 from bacterial flagellin by the receptor kinase FLS2. The authors observed that DSF pre-treatment inhibits several immune responses such as production of reactive oxygen species, stomata closure, and accumulation of callose. The authors then propose that plant plasma membrane lipidome remodeling induced by DSF alters FLS2 nanoscale organization at the plasma membrane, delays ligand-induced FLS2 endocytosis, and causes defect in immune responses previously mentioned. Our understanding of the mechanisms regulating cell surface receptors nanoscale organization and their potential importance for signaling is extremely limited. In that sense, the observations made by Tran and co-authors are of special interest.

However, I would encourage the authors to reformulate some aspects to their manuscript, so that it is ultimately more in line with some of their own data and that of previously published studies. Specific comments.

1. The authors propose that defects observed upon DSF treatment are caused by an over-accumulation in sterol species (Figure 3D). It should be noted that DSF alleviates several defects induced by M β CD treatment [e.g. on FLS2 endocytosis (Figure 4A, Figure S5), flg22-induced ROS burst (Figure 4C), REM1.2 nanodomain organization (Figure 5A) and FLS2 fluorescence anisotropy (Figure 5D)] without actually alleviating the decrease in sterol species induced by M β CD treatment (Figure 3F). This would therefore suggest that the increase in sterol species observed after DSF treatment does not underlie the effect of DSF observed on these processes. In contrary, DSF does not alleviate M β CD effect on primary root length (Figure 3E), which may indicate that DSF's effect on primary root growth is linked to an over-accumulation of sterol species. In good agreement, effect of DSF on primary root growth is abolished when sterol biosynthesis is altered (Figure 3G). The authors should consider that several mechanisms are probably responsible for the different phenotype observed, and should at least discuss these.

2. The authors propose that a potential defect in FLS2 homo-dimerisation explains why production of reactive oxygen species but not phosphorylation of MAPK is affected by DSF [Figure 2E, Figure S2D, E, Figure 5C,D and associated text (L184-204; 299-300; 390-395)]. In vitro studies such as gel filtration assays and structural crystallography however argued since the original Plant Cell paper cited that FLS2 does not associate with itself (Sun et al., Science 2013). Also, using BiFC and FRET in vivo, Ali and co-authors showed that FLS2 does not homodimerize either constitutively or in the presence of flg22 (Ali et al., Plant Cell Physiology 2007). Similarly, using FRET-ABP approach, Somssich and co-authors did not detect direct association of FLS2 with itself (Somssich et al., Science Signaling 2015). The authors should note that in Sun et al., Plant Cell 2012, FLS2 self-association have been detected by co-immunoprecipitation suggesting close proximity of FLS2 molecules but not necessarily physical interaction between them. Co-immunoprecipitation results can be explained by the fact that FLS2 is segregated into nanodomains, which likely impose close proximity of FLS2 molecules but not their direct intermolecular interaction. Moreover, there is no indication that proximity of FLS2 molecules, as measured by co-immunoprecipitation in Sun et al., Plant Cell 2012, is enhanced upon flg22 perception nor that it influences FLS2 phosphorylation and/or signal transduction. While so far there is no indication that FLS2 physically associates with itself and therefore no indication that FLS2 self-association regulates signaling triggered by flg22, many mechanisms regulating RBOHD activity have been uncovered in the past years (e.g. Kadota et al., Mol. Cell 2014; Li et al., Cell Host Microbe 2014; Monaghan et al., Cell Host Microbe 2014 ; Liang et al., eLife 2016; Wang et al., Mol. Cell 2018; Liang et al., Cell Res. 2018). In addition, RBOHD

function have been proposed to be regulated in sterol-enriched membrane domains in tobacco and Arabidopsis (e.g. Liu et al., Plant J. 2009; Lherminier et al., Mol. Plant Microb. Inter. 2009; Simon-Plas et al., Trends Plant Sci. 2011; Hao et al., Plant Cell 2014). Thus, as mentioned in the discussion, it is conceivable that DSF may alter one or several of these processes. The results section and rationale presented by the authors should better reflect this.

Finally, how FLS2 self-association parameters have been determined from fluorescence anisotropy values and with which certainty? Could subtle, yet apparently significant, decrease of FLS2-GFP fluorescence anisotropy be due to changes in FLS2 membrane environment (Keinath et al., JBC 2010), spatio-temporal arrest (Ali et al., Plant Cell Phys. 2007; Bürcherl et al., eLife 2017), and perhaps increased nano-clustering following flg22 perception rather than FLS2 homo-oligomerization? Of note, fluctuation in fluorescence anisotropy is often interpreted as variation in molecules nano-clustering rather than direct protein-protein interaction (for example Raghupathy et al., Cell 2015), and should probably be interpreted as such here.

3. The authors suggest that delayed endocytosis of FLS2 underlies defect in ROS production observed upon DSF treatment. The authors should note that previous study showed that impaired FLS2 endocytosis is linked to an increase in ROS production (e.g. Spallek et al., PLoS Genetics 2013; Cui et al., development 2018). In lines 137-138 and 161-172, it should be noted that endocytic internalization of FLS2 does not activate innate immunity. Instead, it has been shown that endocytosis and degradation of FLS2 occurs after activation of early immune signaling, and thus is most likely part of a post-activation degradation mechanisms to deplete receptors from the plasma membrane and give space to de novo synthesized ligand-free receptors (Lu et al., Science 2011; Beck et al., TPC 2012; Spallek et al., PLoS Genetics 2013; Mbengue et al., PNAS 2016). Please correct accordingly.

4. Figure S5B and C: Results for M β CD and lovastatin treatments are in contradiction with previous results reported by Cui et al., Development 2018, where an increase in flg22-induced ROS burst has been observed in *smt1* mutant and upon M β CD treatment. How do the authors explain this discrepancy?

5. Figure 2B: When combined, DSF and flg22 seem to reduce signal observed for FLS2-GFP. This is not discussed by the authors, and seems counterintuitive as DSF is proposed to limit flg22-induced FLS2 endocytosis (Figure 2A).

6. Figure S1F and related text (L169-171): Overexpression of FLS2 and BAK1 in protoplasts is not optimal to study potential defect in flg22-induced FLS2-BAK1 complex formation. For example, implication of LORELEI (LRE)-LIKE GLYCOSYLPHOSPHATIDYLINOSITOL (GPI)-ANCHORED PROTEIN 1 (LLG1) in FLS2-BAK1 complex formation could be observed for native FLS2 and BAK1 in seedlings (Xiao et al., Nature 2019) but not in protoplast overexpressing FLS2 and BAK1 (Shen et al., PNAS 2017). Therefore, the authors should be more cautious with the conclusions drawn from these experiments.

7. L299: To conclude that M β CD treatment alleviates the effect of DSF on FLS2-GFP fluorescence anisotropy the authors should have compared DSF alone versus DSF+M β CD side-by-side.

8. Figure 4B: To be able to interpret this experiment, the authors should have compared the effect of DSF with or without M β CD and ES9-17.

9. Figure 5A: It should be noted that DSF does not phenocopy the addition of sterol here while the effect of DSF is proposed to be due to an increase in the amount of sterols.

10. L107-112: Inhibition of root growth by a molecule that derives from plant pathogen usually indicate activation of PTI not a dysfunction of it. Therefore, inhibition of root growth induced by DSF should not be interpreted as a consequence of DSF inhibition of PTI. Please rephrase.

11. Figure S1D: To be analyzed and interpreted, flg22-induced MAPK phosphorylation samples for DMSO and DSF treatment should be presented side-by-side on the same membrane.

12. The authors state that osmotic stress rescues the inhibition of endocytosis caused by DSF. The authors do not present data concerning FLS2 endocytosis upon treatment with mannitol.

- Rescue of FLS2 endocytosis might be specifically observed upon salt stress and not osmotic stress per se. Is salt treatment sufficient to trigger FLS2 endocytosis? I would suggest removing this part from the manuscript as it is not central to their study.
13. L302-304: There is no experimental evidence directly linking biophysical properties of the plasma membrane and endocytosis in plants. Please rephrase.
 14. Related to Figure 1C, Figure 4C and Figure S5B-C: the authors should provide statistical analysis of the flg22-induced ROS burst experiments. For example, the authors could plot and compare total photon counts detected for each condition.
 15. L166-168: Please rephrase, DSF does not have significant effect on the accumulation of FRK1 transcripts induced by flg22 perception. Also, replace DSF signaling by DSF treatment (L315).
 16. Figure 1E: Despite the explanation in the material and methods section, calculation of the flg22 protection index remains obscure to me. Could the authors provide values of log₁₀ (CFU/mg) relative to control (DMSO and DSF) to show the effect of flg22 treatment in DMSO and DSF treated plants?
 17. L141-142, Figure 2A: No time-course quantification of FLS2-GFP accumulation at the plasma membrane is provided. Therefore, it is misleading that the authors state that they observed a replenishment of FLS2-GFP back to the plasma membrane. The authors observed a disappearance of flg22-induced endocytic vesicles 120 min after treatment. Please rephrase.
 18. Statistical analysis are missing in Figure S1E.
 19. L150: Measurement of the number of endosomes after flg22 treatment (Figure 1A) is also a quantitative measurement of FLS2-GFP endocytosis. Please rephrase.
 20. L161-162: The authors analyzed FLS2 endocytosis and degradation not FLS2 recycling. Please rephrase.
 21. L136-137: flg22 does not induced conformational changes in FLS2 receptor (Sun et al., Science 2013). Please rephrase and note that reference 22 is not relevant here.
 22. L140: the authors should explain to the readers what is flgII-28, and why they used it.
 23. 182-183: Reference 39 does not refer to FLS2 nanodomains, please correct.
 24. Figure 2E: It seems surprising that flg22 and DSF+flg22 condition are significantly different.
 25. L230-231 and L238-239: It would be of interest to refer to publications reporting the implication of lipids in regulating membrane compartmentalization and function in plant (e.g. Grison et al., Plant Cell 2015; Gronnier et al., eLife 2017; Platre et al., Science 2019; Huang et al., PNAS 2020).
 26. Figure 4A: Please replace sterol by DSF above the bar graph.
 27. L292: Change flexibility by biophysical properties (membrane flexibility has not been studied in Grosjean et al., JBC 2015)
 28. For consistency, change microdomain into nanodomain through the text (e.g L261, L297, 302, 335, 397 etc.)
 29. L389-390: references 33 and 102 do not document the effect of disrupted plasma membrane continuity on BRI1 and FLS2 signaling. Please rephrase.
 30. L308-309: It is surprising to me that NaCl, which is known to limit root growth, restored root growth when combined with DSF.
 31. Figure 1. Replace against by induced or stimulated by the bacterial PAMP flg22
 32. The number of experiments performed for each panel and number of cells analyzed for each experiment should be indicated in the figure legends.
 33. L397-399: I wonder what the link between FLS2 endocytosis, FLS2 nano-clustering, ROS production and reference 104 is.
 34. Line 425: As TLR5 has no phylogenetic relationship with FLS2, it cannot be referred to as an FLS2 ortholog.
 35. The authors should cite previous publications having already shown that ROS production and MAPK activation are uncoupled (Segonzac et al., Plant Physiol 2011; Ranf et al., Plant J 2011; Xu et al., Plant J 2014).

Reviewer #2 (Comments to the Authors (Required)):

In this study, the authors report that DSF QS molecule from pathogenic bacteria, such as *Xanthomonas campestris* pv *campestris*, induces sterol production in *Arabidopsis* to suppress the receptor clustering and endocytosis in flg22-triggered immune responses. DSF alteration of lipid profiles, in particular an increase in phytosterol species, impairs clathrin-mediated endocytosis that is required for FLS2 immune function. How QS molecules influence host pathology and immunity represents an emerging interesting topic in both plants and animals. Therefore, this work is timely and provide novel insight, which can appeal to a broad readership of the journal, if the main conclusions are validated.

Although I see the potential significance of the work, I have two major concerns that the authors have not sufficiently addressed.

1) DSF was previously reported to activate immune responses in plants, including *Arabidopsis*. In this study, apparent discrepancy regarding DSFs' immune-stimulatory and immune-suppressive functions remains to be reconciled or carefully addressed in experimentation or discussion. It may reflect possible differences in the DSF dose optima, as mentioned in the text, but callose deposition is induced in response to DSF at 20 μ M or higher in *Arabidopsis* (Ref 11), while several FLS2 outputs are suppressed at 25 μ M in this study. DSF dose dependence needs to be examined at least for some defense output(s), in addition to root growth (Figure S1).

If immune-stimulatory effects are not detected under the present conditions, discrepancy needs to be mentioned and possible causes be discussed. If they are detected, it is important to test whether they are also dependent on DSF-induced sterol production.

2) Given the pleiotropic effects of DSF, I wonder how specific their observed DSF effects are to FLS2 immunity or PTI. Although uncoupling FLS2-BAK1 complex formation and MAPK activation from ROS burst, stomatal closure, callose deposition and bacterial resistance is very interesting, DSF broadly impairs CME-mediated endocytosis of BRI1 and BOR1, and likely more. Moreover, DSF effects were assessed after 24-h pretreatment, but such a long exposure to DSF could increase its pleiotropic effects.

Validation of key conclusions following DSF treatment for a minimal exposure time may help to minimize pleiotropic effects.

To strengthen the physiological relevance of the observed DSF effects, it seems important to test whether and if so to which extent DSF-induced alterations in lipid profiles, at least an increase in phytosterol contents, occur during bacterial infection in a DSF-dependent manner, e.g. by comparing inoculation effects between DSF-producing and -deficient bacteria.

Detailed concerns.

3) It is important and feasible to test whether DSF inhibition is specific to flg22 responses or effective against different PAMPs. If it is not seen with another PAMP(s), it could argue against my concern above.

- 4) In Figure 2D, DSF induces a significant decrease in SCI, with or without flg22, while flg22 does not alter SCI. Could the authors comment on the DSF effects and the lack for detectable changes in response to flg22? If flg22 effects are not significant in SCI analysis, the discussion in Line 205 "consistent with SCI analysis" seems not logical.
- 5) In Figure S6, why did the authors show recovery of FLS2 internalization at 100 mM NaCl, but not at 10 or 50 mM, at which root growth is recovered?
- 6) Line 318, wording "symbionts" is not suitable for *Pseudomonas syringae* or *Xanthomonas* pathogens.
- 7) Lines 350-351, DSF inhibition of PTI seems to be prominent at a late stage of infection. The possible relevance of PTI suppression at this stage should be discussed. It would help if the authors refer to PTI-phasing (Lu et al, PNAS 2009; Tsuda et al, PLoS genetics 2009) and PTI-ETI mutual potentiation papers (bioRxiv <https://doi.org/10.1101/2020.04.10.034173> <https://doi.org/10.1101/2020.04.10.031294>).
- 8) In Figure 6, a magnified picture of the lipid-bilayer PM should be shown closer to the PM, instead of the bottom.
- 9) Line 425, because TLR5 has a different module structure compared to FLS2 and recognizes a distinct epitope from flg22, they are not orthologous or even homologous to each other, in terms of the receptor structure and function.

First of all, we would like to express our appreciation for the chance to revise and re-submit our manuscript to Life Science Alliance. We also want to thank the reviewers for taking their time to go through our manuscript and provided us with a lot of useful feedback, which helped us to improve the quality of the manuscript significantly. We agree with the reviewers that the manuscript can benefit from a more thorough discussion of our results and presentation of the data. We, therefore, have revised our manuscripts to present our results more clearly, integrated some of the changes and experiments requested by the reviewers and included relevant references that support our conclusion and discussion. At the same time, we also thank the reviewers for appreciating the novelty and importance of our manuscript. In this revised version, we would like to address several concerns raised by the two reviewers point-by-point as follows:

Reviewer #1 (Comments to the Authors (Required)):

In their manuscript, Tran and co-authors study the effect of a quorum sensing molecule called diffusible-signal factor (DSF) on plant immune responses triggered by the perception of the epitope flg22 from bacterial flagellin by the receptor kinase FLS2. The authors observed that DSF pre-treatment inhibits several immune responses such as production of reactive oxygen species, stomata closure, and accumulation of callose. The authors then propose that plant plasma membrane lipidome remodeling induced by DSF alters FLS2 nanoscale organization at the plasma membrane, delays ligand-induced FLS2 endocytosis, and causes defect in immune responses previously mentioned. Our understanding of the mechanisms regulating cell surface receptors nanoscale organization and their potential importance for signaling is extremely limited. In that sense, the observations made by Tran and co-authors are of special interest.

However, I would encourage the authors to reformulate some aspects to their manuscript, so that it is ultimately more in line with some of their own data and that of previously published studies.

Specific comments.

1. The authors propose that defects observed upon DSF treatment are caused by an over-accumulation in sterol species (Figure 3D). It should be noted that DSF alleviates several defects induced by M β CD treatment [e.g. on FLS2 endocytosis (Figure 4A, Figure S5), flg22-induced ROS burst (Figure 4C), REM1.2 nanodomain organization (Figure 5A) and FLS2 fluorescence anisotropy (Figure 5D)] without actually alleviating the decrease in sterol species induced by M β CD treatment (Figure 3F). This would therefore suggest that the increase in sterol species observed after DSF treatment does not underlie the effect of DSF

observed on these processes. In contrary, DSF does not alleviate M β CD effect on primary root length (Figure 3E), which may indicate that DSF's effect on primary root growth is linked to an over-accumulation of sterol species. In good agreement, effect of DSF on primary root growth is abolished when sterol biosynthesis is altered (Figure 3G). The authors should consider that several mechanisms are probably responsible for the different phenotype observed, and should at least discuss these.

We thank the reviewer for the comment. M β CD is a well-established chemical molecule that is used to deplete sterol species from the cells acutely. Hence, the sterol-depletion effect by M β CD is theoretically dominant over most of the cellular regulatory mechanisms in changing sterols, including the impact of DSF on sterol accumulation (Figure 3F). However, DSF still counteracts with M β CD in changing sterol content, though being a weaker opponent. This is supported by the fact that while DSF alone could induce up to 165% increase in total sterol content compared to DMSO control, adding M β CD together with DSF didn't simply reduce the sterol accumulation but caused a decrease of 45% in total sterol compared to DMSO treatment. We now included another sentence to explain the differences in our experimental findings (L253-256).

2. The authors propose that a potential defect in FLS2 homo-dimerisation explains why production of reactive oxygen species but not phosphorylation of MAPK is affected by DSF [Figure 2E, Figure S2D, E, Figure 5C,D and associated text (L184-204; 299-300; 390-395)]. **In vitro studies such as gel filtration assays and structural crystallography however argued since the original Plant Cell paper cited that FLS2 does not associate with itself (Sun et al., Science 2013).** Also, using BiFC and FRET in vivo, **Ali and co-authors showed that FLS2 does not homodimerize either constitutively or in the presence of flg22 (Ali et al., Plant Cell Physiology 2007).** Similarly, using FRET-ABP approach, Somssich and co-authors did not detect direct association of FLS2 with itself (Somssich et al., Science Signaling 2015). The authors should note that in **Sun et al., Plant Cell 2012, FLS2 self-association have been detected by co-immunoprecipitation suggesting close proximity of FLS2 molecules but not necessarily physical interaction between them.** Co-immunoprecipitation results can be explained by the fact that FLS2 is segregated into nanodomains, which likely impose close proximity of FLS2 molecules but not their direct intermolecular interaction. Moreover, there is **no indication that proximity of FLS2 molecules, as measured by co-immunoprecipitation in Sun et al., Plant Cell 2012, is enhanced upon flg22 perception nor that it influences FLS2 phosphorylation and/or signal transduction.** While so far there is no indication that FLS2 physically associates with itself and therefore no indication that FLS2 self-association regulates signaling triggered by flg22, many mechanisms regulating RBOHD activity have been uncovered in the past years (e.g. Kadota et al., Mol. Cell 2014; Li et al., Cell Host Microbe 2014; Monaghan et al., Cell Host Microbe 2014 ; Liang et al., eLife 2016; Wang et al., Mol. Cell 2018; Liang et al., Cell Res. 2018). **In addition, RBOHD function have been proposed to be regulated in sterol-enriched membrane domains in tobacco and Arabidopsis (e.g. Liu et al., Plant J. 2009; Lherminier et al., Mol. Plant Microb. Inter. 2009; Simon-Plas et al., Trends Plant Sci. 2011; Hao et al., Plant Cell 2014).** **Thus, as mentioned in the discussion, it is conceivable that DSF may alter one or several of these processes. The results section and rationale presented by the authors should better reflect this.**

We thank the reviewer for the thoughtful comment. And we agree that we should conclude and discuss the physical FLS2 interaction from our results with more caution to avoid

potential misinterpretation. Sun et al. 2013 showed the structure of **FLS2-LRR** and **BAK1-LRR domains** (not full-length FLS2 and BAK1) as a monomeric heterodimer induced by flg22. However, in the same paper, they also acknowledged that it is possible that **full-length FLS2** could form homo-oligomers. Sun et al. 2012 Plant Cell actually showed that phosphorylation sites of FLS2 are required for FLS2-FLS2 association, and this FLS2-FLS2 association increased over time after flg22 elicitation (although no evidence was shown to prove that this association influences FLS2 phosphorylation as the reviewer pointed out, and we have removed this sentence at line 210 to avoid confusion). As suggested by Sun et al. 2012, the negative data regarding FLS2-FLS2 interaction observed by Ali et al. (2007) could be due to either the overexpression of FLS2 with fluorescence tags in protoplast not mimicking the native FLS2 protein, or an intermediary bridging molecule between FLS2 molecules exists, rather than the failure of FLS2 to form oligomers (although we could not judge the FRET result in Ali et al. 2007 as well as the resolution/sensitivity of the microscopic techniques 13 years ago since we could not identify the technical information about the sensitivity of the camera provided in the method). The model suggested by Somssich et al. 2015 Science Signaling supported this later point as they suggested the existence of a tetrameric complex of FLS2 and BAK1.

Our possible explanation for the homo-FRET result could be that the shifts in anisotropy after flg22 treatment was due to spatio-temporal arrest of FLS2 molecules at PM, which could trigger the formation of FLS2 nano-clusters. FLS2 nano-clusters due to spatio-temporal arrest at PM could activate directly or indirectly PTI signalling by modulating its interacting partners, thus innate immunity cascades (L392-395).

To avoid further dispute regarding the oligomerization state of FLS2 from the different results in the literature, we will now rephrase “FLS2-oligomerization” as “FLS2 nano-clustering”/“FLS2 clustering at nano-scale” throughout the paper.

In addition, we appreciate the reviewer pointing out other possible mechanisms that DSF can affect plant innate immunity, such as RBOHD function. We did mention this possibility briefly in our discussion, but we now reflect this in the model in Figure 6 and included the additional references that the reviewer suggested (line 374-378).

Finally, how FLS2 self-association parameters have been determined from fluorescence anisotropy values and with which certainty? Could subtle, yet apparently significant, decrease of FLS2-GFP fluorescence anisotropy be due to **changes in FLS2 membrane environment** (Keinath et al., JBC 2010), **spatio-temporal arrest** (Ali et al., Plant Cell Phys. 2007; Bürcherl et al., eLife 2017), and perhaps **increased nano-clustering following flg22 perception** rather than FLS2 homo-oligomerization? Of note, fluctuation in fluorescence anisotropy is often interpreted as variation in molecules nano-clustering rather than direct protein-protein interaction (for example Raghupathy et al., Cell 2015), and should probably be interpreted as such here.

We agree that there is a possibility that a decrease of FLS2-GFP fluorescence anisotropy could be due to change in FLS2 membrane environment, in that case, intensity profile of nano-scale (< 10 nm) homo-FRET signals should follow concentration dependence. Our experiments showed intensity independent homo-FRET signals (please see the below graphs as examples).

Therefore, the possibility of FLS2-nanoclustering due to spatio-temporal arrest is a more likely explanation, and we agree with the reviewer in that context. Thank you very much for your detailed explanation. We did interpret that the changes in anisotropy should be as molecular nano-clustering, and instead of referring to this phenomenon, we now replaced “FLS2-FLS2 homo-oligomerization” as “FLS2 clustering ”/”FLS2 nano-clustering” to avoid such confusion with protein physical interaction.

Figure removed by Life Science Alliance Editorial Staff per authors' request

3. The authors suggest that delayed endocytosis of FLS2 underlies defect in ROS production observed upon DSF treatment. The authors should note that previous study showed that impaired FLS2 endocytosis is linked to an increase in ROS production (e.g. Spallek et al., PLoS Genetics 2013; Cui et al., development 2018). In lines 137-138 and 161-172, it should be noted that endocytic internalization of FLS2 does not activate innate immunity. Instead, it has been shown that endocytosis and degradation of FLS2 occurs after activation of early immune signaling, and thus is most likely part of a post-activation degradation mechanisms to deplete receptors from the plasma membrane and give space to de novo synthesized ligand-free receptors (Lu et al., Science 2011; Beck et al., TPC 2012; Spallek et al., PLoS Genetics 2013; Mbengue et al., PNAS 2016). Please correct accordingly.

We thank the reviewer for the comment and agree that FLS2 endocytic internalization does not activate innate immunity. We have now rephrased this sentence to make it clearer: “Flagellin binding causes the rapid association of the FLS2 receptor with its interacting

partners to activate the innate immunity cascade and subsequently leads to the endocytic internalization of the receptor” (L140-142).

To avoid causing the confusion that receptor endocytosis leads activation of innate immunity, we also revised the sentence at line 161-172 (previous version) to “We did not observe a noticeable change in the acute response of MAPK phosphorylation, which occurs approximately 15 min post-elicitation and returns to basal level within 60 min.” (L168). In addition, we also added a sentence (L374-378) to discuss that RBOHD function itself could also be affected by DSF perturbation to avoid the suggestion that FLS2 endocytosis underlies the defect in ROS production.

4. Figure S5B and C: Results for M β CD and lovastatin treatments are in contradiction with previous results reported by Cui et al., Development 2018, where an increase in flg22-induced ROS burst has been observed in *smt1* mutant and upon M β CD treatment. How do the authors explain this discrepancy?

Regarding the discrepancy, we are aware that Cui et al. reported M β CD treatment caused an increase in ROS burst (Figure S1) compared to mock treatment while we showed in our study that ROS was reduced with M β CD treatment. The differences in ROS assays conducted in our study vs. Cui et al. 2018 could have stemmed from the fact that the experimental conditions were different: we used flg22 at 1 μ M and M β CD at 2mM, whereas in Cui et al., 100 nM flg22 and 10mM M β CD (30 min) were used. We suspect that the increased ROS caused by 10mM M β CD reported by Cui et al. 2018 could be a result of a strong disruption of lipid compartmentalization and RBOHD behaviors on the cell surface, although this speculation needs to be further investigated in future studies. We now have discussed this difference in the discussion (Line 380-383).

Regarding the increased ROS of *smt1-1* mutant, we would like not to comment on Cui et al. 2018 result, as we observed certain growth defect in this mutant. Without knowing the detailed mechanisms that might perturb the ROS system, we chose not to proceed further with other functional assays beyond the growth experiment.

5. Figure 2B: When combined, DSF and flg22 seem to reduce signal observed for FLS2-GFP. This is not discussed by the authors, and seems counterintuitive as DSF is proposed to limit flg22-induced FLS2 endocytosis (Figure 2A).

The reduction in signal from DSF+FLS2 treatment could be the result of the much-decreased clustering of FLS2 at the same treatment condition. 1. DSF attenuates the FLS2-clustering; 2. DSF does not completely block endocytosis at 60 min (Figure 2A), which resulted in the internalization of the effective FLS2 receptors. 3. The additive effect from the above two reasons will drastically reduce the brightest spots from the cell surface. Due to the quantum yield bias with lower absorption signals (less-clustered foci), the detectable photon under particular imaging condition could be shifted by a factor of 2 or more between clustered (brightest foci) vs. diffused receptor (weak signals) (1). Due to such a challenge in providing an accurate quantitative comparison between the images with high contrast in intensity, especially for such weak fluorescent FLS2-GFP, we prefer not to interpret this point too much to avoid potentially misleading.

6. Figure S1F and related text (L169-171): Overexpression of FLS2 and BAK1 in protoplasts is not optimal to study potential defect in flg22-induced FLS2-BAK1 complex formation. For example, implication of LORELEI (LRE)-LIKE GLYCOSYLPHOSPHATIDYLINOSITOL (GPI)-ANCHORED PROTEIN 1 (LLG1) in FLS2-BAK1 complex formation could be observed for native FLS2 and BAK1 in seedlings (Xiao et al., Nature 2019) but not in protoplast overexpressing FLS2 and BAK1 (Shen et al., PNAS 2017). Therefore, the authors should be more cautious with the conclusions drawn from these experiments.

We thank the reviewer for the comment regarding the effect of protoplast overexpression on protein interaction. Here, we only made our conclusion solely on the FLS2-BAK1 interaction, and we have rephrased the sentence (L190-194) to refer to this result as “FLS2-BAK1 association” instead of “FLS2-BAK1 complex” to avoid making any assumption about the FLS2-BAK1 complex which may contain other proteins that can be affected by overexpression in protoplast.

7. L299: To conclude that M β CD treatment alleviates the effect of DSF on FLS2-GFP fluorescence anisotropy the authors should have compared DSF alone versus DSF+M β CD side-by-side.

This is about the homo-FRET experiment. All the MbCD and DSF+MbCD were performed at the same time with DMSO/DSF set (presented in Figure 2E and Figure 5D). As the reviewer suggested, we now added another graph in Fig 5D to show the Anisotropy ratio [flg22/no flg22] for each treatment.

8. Figure 4B: To be able to interpret this experiment, the authors should have compared the effect of DSF with or without M β CD and ES9-17.

We want to note that the data for DSF sets (+/- ES9-17) was already presented in Figure 3A, C. Therefore, we have generated a new graph to compare the ratio of Intracellular/PM signal intensity between ES9-17 vs. control treatment for the DSF- and DSF+MbCD- treated plants (Figure 4B). This notion is also updated in the figure legend of Fig. 4B.

9. Figure 5A: It should be noted that DSF does not phenocopy the addition of sterol here while the effect of DSF is proposed to be due to an increase in the amount of sterols.

We thank the reviewer for pointing this out. This could be the result of the strength-dependency of DSF and sterol treatment in this particular read-out, as sterol may exert a stronger effect on microdomain clustering than DSF, and we were careful not to conclude as such way (DSF phenocopied sterol effect) in this section.

10. L107-112: Inhibition of root growth by a molecule that derives from plant pathogen usually indicate activation of PTI not a dysfunction of it. Therefore, inhibition of root growth induced by DSF should not be interpreted as a consequence of DSF inhibition of PTI. Please rephrase.

We agree with the reviewer that inhibition of plant growth does not necessarily reflect the dysfunction of PTI. To avoid confusion, we rephrased the sentence to “We asked whether DSF, a recently discovered QS signal produced by diverse Gram-negative pathogens, could dysregulate plant growth and pattern-triggered immunity (PTI) responses” (L108)

11. Figure S1D: To be analyzed and interpreted, flg22-induced MAPK phosphorylation samples for DMSO and DSF treatment should be presented side-by-side on the same membrane.

We have now replaced the MAPK phosphorylation experiment with another blot showing the DMSO and DSF treatment in the same membrane, as the reviewer suggested with a similar result in the new Figure S2B.

12. The authors state that osmotic stress rescues the inhibition of endocytosis caused by DSF. The authors do not present data concerning FLS2 endocytosis upon treatment with mannitol. Rescue of FLS2 endocytosis might be specifically observed upon salt stress and not osmotic stress per se. Is salt treatment sufficient to trigger FLS2 endocytosis? I would suggest removing this part from the manuscript as it is not central to their study.

The salt treatment itself does not trigger FLS2 endocytosis. We thank the reviewer for the suggestion, and we have removed this experiment as it is not essential to the main results.

13. L302-304: There is no experimental evidence directly linking biophysical properties of the plasma membrane and endocytosis in plants. Please rephrase.

L302-304 We cited Ref 53 which mentioned how physical cues such as salinity stress possibly regulate a clathrin-independent process across plant roots. However, we now removed this section (salt-induced reversion of DSF-induced inhibition of endocytosis) as suggested by the reviewer in another comment #12 as this experiment is not essential to the core result of the paper.

14. Related to Figure 1C, Figure 4C and Figure S5B-C: the authors should provide statistical analysis of the flg22-induced ROS burst experiments. For example, the authors could plot and compare total photon counts detected for each condition.

We thank the reviewer for the suggestion. We now included the statistics of the total photon count for each ROS experiment (Fig 1C, Fig 4C, Fig S1C-D, Fig S5B-C).

15. L166-168: Please rephrase, DSF does not have significant effect on the accumulation of FRK1 transcripts induced by flg22 perception. Also, replace DSF signaling by DSF treatment (L315).

We now have rephrased this sentence and also replaced “DSF-signaling” with “DSF treatment” throughout the manuscript, as the reviewer suggested.

16. Figure 1E: Despite the explanation in the material and methods section, calculation of the flg22 protection index remains obscure to me. Could the authors provide values of log₁₀ (CFU/mg) relative to control (DMSO and DSF) to show the effect of flg22 treatment in DMSO and DSF treated plants?

Since the protection index is not very intuitive to follow, we now replaced it with the relative ratio of $\text{Log}_{10}(\text{CFU}/\text{mg})$ flg22/control for each of the drug treatment (DMSO or DSF) as the reviewer suggested. The figure legend was also corrected accordingly.

17. L141-142, Figure 2A: No time-course quantification of FLS2-GFP accumulation at the plasma membrane is provided. Therefore, it is misleading that the authors state that they observed a replenishment of FLS2-GFP back to the plasma membrane. The authors observed a disappearance of flg22-induced endocytic vesicles 120 min after treatment. Please rephrase.

We thank the reviewer for pointing this out. We have now corrected the sentence to “Upon treatment of flagellin peptide flg22 but not flgII-28, FLS2 endocytic internalisation was evidenced by the appearance of punctate endosomes after around 60-75 min and disappeared at 120 min post elicitation”. (L143-147)

18. Statistical analysis are missing in Figure S1E.

We now added the statistical analysis for *FRK1* expression for this figure panel.

19. L150: Measurement of the number of endosomes after flg22 treatment (Figure 1A) is also a quantitative measurement of FLS2-GFP endocytosis. Please rephrase.

We thank the reviewer for the correction. We now rephrase to the sentence to “To further confirm the DSF-caused defects in receptor endocytosis. We examined the FLS2 endocytic internalization at the plasma membrane using Variable-angle epifluorescence microscopy (VAEM)”. (L156-159)

20. L161-162: The authors analyzed FLS2 endocytosis and degradation not FLS2 recycling. Please rephrase.

We thank the reviewer for the correction, as suggested here and as FLS2 endocytosis & degradation is not related to MAPK response, we now revised this sentence to “We did not observe a noticeable change in the acute response of MAPK phosphorylation...” (L167-168)

21. L136-137: flg22 does not induced conformational changes in FLS2 receptor (Sun et al., Science 2013). Please rephrase and note that reference 22 is not relevant here.

We thank the reviewer for the correction. We now rephrase the sentence (L141-143) to “Flagellin binding causes the rapid association of the FLS2 receptor with its interacting partners to activate the innate immunity cascade and subsequently lead to the endocytic internalization of the receptor”. We moved ref 22 (showing BIK1 association with FLS2-BAK1) to the end of the sentence as it is relevant to the point that FLS2 associates with other proteins upon flagellin binding.

22. L140: the authors should explain to the readers what is flgII-28, and why they used it.

We now revise the sentence to “...but not flgII-28 (a flagellin epitope that Arabidopsis does not recognize, as a negative control)” (L143-147)

23. 182-183: Reference 39 does not refer to FLS2 nanodomains, please correct.

182-183 FLS2 formed heterogeneous PM clusters with or without ligand activation (Fig. 2B), representing the resting- or activated- states, respectively (39).

We now rephrased this reference to avoid confusion (L189-190). We thank the reviewer for the correction.

24. Figure 2E: It seems surprising that flg22 and DSF+flg22 condition are significantly different.

The DSF+flg22 condition has a clear shift of distribution, although the mean values between DSF and DSF+flg22 may look close. The DSF+flg22 condition has a much-reduced population of data with low anisotropy signal. We plot anisotropy value here on the histogram to show the changes in the distribution of anisotropy values in different treatments.

Figure removed by Life Science Alliance Editorial Staff per authors' request

25. L230-231 and L238-239: It would be of interest to refer to publications reporting the implication of lipids in regulating membrane compartmentalization and function in plant (e.g. Grison et al., Plant Cell 2015; Gronnier et al., eLife 2017; Platre et al., Science 2019; Huang et al., PNAS 2020).

We now included these references in the sentence at L232-234 to support this point. We thank the reviewer for the suggestion.

26. Figure 4A: Please replace sterol by DSF above the bar graph.

We appreciate the reviewer for pointing out this error.

The graph in Fig 4A was incorrect as it is from the Sterol+MbCD experiment and was mistakenly inserted there. We now replaced the graph with the correct DSF+MbCD graph.

27. L292: Change flexibility by biophysical properties (membrane flexibility has not been studied in Grosjean et al., JBC 2015)

We thank the reviewer for the correction. We now rephrased this sentence to “Sterols are constituents of the liquid-ordered lipid nanodomains and influence membrane organization (L295-296)

28. For consistency, change microdomain into nanodomain through the text (e.g L261, L297, 302, 335, 397 etc.)

We now replace “microdomain” with “nanodomain” to make it consistent throughout the manuscript.

29. L389-390: references 33 and 102 do not document the effect of disrupted plasma membrane continuity on BRI1 and FLS2 signaling. Please rephrase.

We thank the reviewer for the correction. We now rephrased the sentence to “disruption of PM continuity is competent enough to alter the clustering behavior of multiple PM receptors.”

30. L308-309: It is surprising to me that NaCl, which is known to limit root growth, restored root growth when combined with DSF.

NaCl was shown to induce endocytosis in Arabidopsis roots (Baral et al. 2015 The Plant Cell). We speculate that the recovery of growth could have been a result of the recovery of endocytosis induced by NaCl. However, as the reviewer suggested in a related comment (#12), we now removed this data as it is not essential to the main results of the paper.

31. Figure 1. Replace against by induced or stimulated by the bacterial PAMP flg22

We now have revised the Figure 1 legend text from “against” to “stimulated by”.

32. The number of experiments performed for each panel and number of cells analyzed for each experiment should be indicated in the figure legends.

We now revised the figure legend text to include the information about the number of experiments and cells analyzed for each experiment. We thank the reviewer for the suggestion.

33. L397-399: I wonder what the link between FLS2 endocytosis, FLS2 nano-clustering, ROS production and reference 104 is.

Ref 104 was supporting our point about how membrane lipids could affect the nano-clustering of membrane proteins. We now revise this sentence to “Our data suggest a plausible model (Fig. 6) in which the Xcc QS molecule DSF alters sterols composition and thereby, modulates the clustering of membrane microdomains. This modulation of membrane affects both the FLS2 nano-clustering upon PAMP stimulation, general endocytosis pathway and ROS production, all of which rely on the integrity and compartmentalization of the plasma membrane lipids”. (Line 397).

34. Line 425: As TLR5 has no phylogenetic relationship with FLS2, it cannot be referred to as an FLS2 ortholog.

We thank the reviewer for the correction. We now removed this sentence to avoid unnecessary confusion in our speculation.

35. The authors should cite previous publications having already shown that ROS production and MAPK activation are uncoupled (Segonzac et al., Plant Physiol 2011; Ranf et al., Plant J 2011; Xu et al., Plant J 2014).

We thank the reviewer for the suggestion. We now included these references in our discussion.

Reviewer #2 (Comments to the Authors (Required)):

In this study, the authors report that DSF QS molecule from pathogenic bacteria, such as *Xanthomonas campestris* pv *campestris*, induces sterol production in *Arabidopsis* to suppress the receptor clustering and endocytosis in flg22-triggered immune responses. DSF alteration of lipid profiles, in particular an increase in phytosterol species, impairs clathrin-mediated endocytosis that is required for FLS2 immune function. How QS molecules influence host pathology and immunity represents an emerging interesting topic in both plants and animals. Therefore, this work is timely and provide novel insight, which can appeal to a broad readership of the journal, if the main conclusions are validated.

Although I see the potential significance of the work, I have two major concerns that the authors have not sufficiently addressed.

1) DSF was previously reported to activate immune responses in plants, including *Arabidopsis*. In this study, apparent discrepancy regarding DSF, immune-stimulatory and immune-suppressive functions remains to be reconciled or carefully addressed in experimentation or discussion. It may reflect possible differences in the DSF dose optima, as mentioned in the text, but callose deposition is induced in response to DSF at 20 μ M or higher in *Arabidopsis* (Ref 11), while several FLS2 outputs are suppressed at 25 μ M in this study. DSF dose dependence needs to be examined at least for some defense output(s), in addition to root growth (Figure S1). **If immune-stimulatory effects are not detected under the present conditions, discrepancy needs to be mentioned and possible causes be discussed. If they are detected, it is important to test whether they are also dependent on DSF-induced sterol production.**

We agree with the reviewer that Reference 11 (Kakkar 2015 JXB) reported that DSF induced callose deposition at 20 μ M. However, this number was from an experiment performed in *N. benthamiana*. In *Arabidopsis*, the authors reported callose deposition at DSF concentrations equal to or higher than 50 μ M. DSF concentrations lower than 50 μ M was not tested for *Arabidopsis* in Ref 11, and the higher DSF concentration of 100 μ M was used exclusively for other experiments (cell death staining, infection assay, etc.). We want to emphasize that what we reported in this work is the ability of DSF to suppress PTI response triggered by PAMPs, which was different from the cell death response triggered by DSF itself (without any ligand) reported by Kakkar et al. 2015. As these are two distinct phenomena, we believe our

observations and those reported in Kakkar et al. 2015 are not directly conflicting with each other.

We acknowledge the reviewer's concern about the dose-dependent response to DSF and the possibility of dose optima, and to answer the reviewer's question about the dose-dependency of DSF-induced responses, we have now included a titration with another assay (flg22-induced ROS production) using the same concentrations of DSF that were used in the growth assay (0, 10, 25, 40 μ M). We showed that a lower concentration of DSF (10 μ M) was not immune-stimulatory and DSF at higher concentrations can inhibit ROS burst (Fig S1C). We also performed a more detailed dose-dependent titration of DSF using callose assay (without any ligand) and showed that only higher concentrations of DSF (30-100 μ M) induced significantly more callose deposition in *Arabidopsis* (Figure S1B). This confirms our previous finding in Fig. 1B that at 25 μ M of DSF, on its own, did not have an immune-stimulatory effect on *Arabidopsis*. Therefore the use of 25 μ M of DSF also avoids potential complications from callose disposition or cell death that were triggered by a higher dose of DSF. We think our results and Kakkar et al. 2015 together also suggest the time-dependent and dose-dependent DSF-effects on host biology to mimic the cumulative effects of DSF during infection. We added such additional points in the discussion now.

2) Given the pleiotropic effects of DSF, I wonder how specific their observed DSF effects are to FLS2 immunity or PTI. Although uncoupling FLS2-BAK1 complex formation and MAPK activation from ROS burst, stomatal closure, callose deposition and bacterial resistance is very interesting, DSF broadly impairs CME-mediated endocytosis of BRI1 and BOR1, and likely more. Moreover, DSF effects were assessed after 24-h pretreatment, but such a long exposure to DSF could increase its pleiotropic effects.

Validation of key conclusions following DSF treatment for a minimal exposure time may help to minimize pleiotropic effects.

We agree with the reviewer that the effect that DSF exerted on plants that we observed in this study could be beyond the effect on FLS2 immunity as DSF affected the general endocytosis pathway. Therefore, we did not explicitly claim that those effects are specific to this particular receptor, but rather, through an overall disruption of membrane environment which in its turn, can alter plasma membrane proteins behaviour. We now include a ROS assay that was performed with elf26 peptide (Fig S1D) to support this point.

Regarding the reviewer's point about the 24-h treatment of DSF that we used in this study, we think in the real infection scenario, DSF is continually being produced by bacteria during the whole course of infection (at least several days for a typical infection assay). Therefore, 24 h treatment or longer would be within the relative range of biological relevance. However, we agree that due to the importance of lipid homeostasis on the cell surface, we cannot exclude the existence of other potential effects of DSF that we have not identified in this study.

In addition, here we address the reviewer's concern by including a time-titration of DSF-treatment on flg22-induced FLS2 endocytosis (Figure S1F). Our result shows a decrease in FLS2-positive endosomes upon flg22 elicitation as the incubation time of pre-treatment with DSF increased, suggesting that the DSF effect was accumulating over time, and that 24 h is

an appropriate time-point to use. We hope this experiment can clarify the reviewer's question.

To strengthen the physiological relevance of the observed DSF effects, it seems important to test whether and if so to which extent DSF-induced alterations in lipid profiles, at least an increase in phytosterol contents, occur during bacterial infection in a DSF-dependent manner, e.g. by comparing inoculation effects between DSF-producing and -deficient bacteria.

We thank the reviewer for the comment. However, as we stated in our method section and also in the result section, the reason we chose to use exogenous application of DSF and not DSF-deficient bacterium was the Type III secretion system activation property of DSF. Although we agree with the reviewer comment that this type of experiment may be more physiologically relevant (using bacterial infection with a DSF-deficient strain), one can foresee the defect in bacterial virulence caused by the inability of the bacterium to produce effector proteins to manipulate host cells, or the failure to form proper biofilm structure (2, 3). This inability to infect plants even when observed, cannot be differentiated from the suppression effect of DSF itself on host defense responses and thus, make it an not an ideal experiment to perform in the first place. In addition, due to the restriction of movement between institutes locally in Singapore and limited access to the lab for our collaborators and for us, we cannot perform the lipidomic experiment with infected plants that the reviewer suggested and we hope the reviewer can understand this situation.

Detailed concerns.

3) It is important and feasible to test whether DSF inhibition is specific to flg22 responses or effective against different PAMPs. If it is not seen with another PAMP(s), it could argue against my concern above.

As comment #1 that the reviewer mentioned above, we now added a ROS assay for elf26, showing that the DSF-induced inhibition of ROS triggered by elf26 is not unique to flg22 (Figure S1C).

References

1. B. van Dam *et al.*, Quantum Yield Bias in Materials With Lower Absorptance. *Physical Review Applied* **12**, 024022 (2019).
2. G. E. Gudesblat, P. S. Torres, A. A. Vojnov, *Xanthomonas campestris* overcomes *Arabidopsis* stomatal innate immunity through a DSF cell-to-cell signal-regulated virulence factor. *Plant physiology* **149**, 1017-1027 (2009).
3. P. S. Torres *et al.*, Controlled synthesis of the DSF cell-cell signal is required for biofilm formation and virulence in *Xanthomonas campestris*. *Environ. Microbiol.* **9**, 2101-2109 (2007).

July 22, 2020

RE: Life Science Alliance Manuscript #LSA-2020-00720R

Prof. Yansong Miao

School of Biological Sciences, Nanyang Technological University, Singapore 637551, Singapore
School of Chemical and Biomedical Engineering, Nanyang Technological University, Singapore 637459, Singapore
60 Nanyang Drive
Singapore 637551
Singapore

Dear Dr. Miao,

Thank you for submitting your revised manuscript entitled "The bacterial QS signal DSF hijacks Arabidopsis sterol biosynthesis to suppress plant immunity". Your work was re-reviewed by one of the original referees whose report is attached below. We would be happy to publish your paper in Life Science Alliance pending final revisions necessary to meet our formatting guidelines.

- Regarding ref 1 points #2 and #24: it would be helpful to include the graphs displayed in response to both points in the SI data of the paper
- ref 1 point #5 and point #9: please include these explanations provided in your response to referees in the revised manuscript.
- ref 1 point #4 consider explaining the discrepancy noted in the referee rebuttal in more detail in the manuscript
- we would encourage you to add scale bars to each panel of Figure S1B
- please take a look at our Manuscript Preparation guidelines and order your manuscript sections accordingly
- please add a callout to Figure S2C
- please provide your manuscript in editable doc format
- please provide your tables in editable doc or excel format
- please use the [10 author names, et al.] format in your references (i.e. limit the author names to the first 10)

A. FINAL FILES:

B. MANUSCRIPT ORGANIZATION AND FORMATTING:

Sincerely,

Reilly Lorenz
Editorial Office Life Science Alliance

Meyerhofstr. 1
69117 Heidelberg, Germany
t +49 6221 8891 414
e contact@life-science-alliance.org
www.life-science-alliance.org

Reviewer #2 (Comments to the Authors (Required)):

I am happy with the revisions made by the authors. There are some points to be addressed.

Lines 381-383. This reads speculative and subjective. Could the authors rephrase this point by giving some details for what remains to be shown in Ref 45 and this study?

Lines 405-410. This logic nicely explains MAPK activation being decoupled from DSF effects on CME and receptor internalization. However, a ROS burst is typically induced earlier, and is not well explained. To comprehend all data, it seems better to predict the existence of another as-yet-unidentified step influencing PRR-NADPH oxidases, rather than simply attributing to time differences between their first detection points.

In Fig 6. FLS2 looks to be homo-dimerized. As discussed with the reviewers, the data available are not strong enough for the occurrence of FLS2 homo-dimerization.

Dear Life Science Alliance Editors,

Thank you for the acceptance of our manuscript (#LSA-2020-00720R) for publication in Life Science Alliance.

We are happy to adopt the following changes in the updated manuscript as suggested by the editor and Reviewer 2 (highlighted in blue font).

-Regarding ref 1 points #2 and #24: it would be helpful to include the graphs displayed in response to both points in the SI data of the paper

We thank the editor for the suggestion and agree that this addition is valuable for the assessment of our results. We now included the graphs mentioned in point #2 and #24 as Figure S2J, K.

-ref 1 point #5 and point #9: please include these explanations provided in your response to referees in the revised manuscript.

The explanations provided in point #5 and 9 are now reflected in the main text at L407-414 and L308-311, respectively.

-ref 1 point #4 consider explaining the discrepancy noted in the referee rebuttal in more detail in the manuscript

We thank the editor for the suggestion. We have now included a more detailed explanation in our discussion (L395-401).

-we would encourage you to add scale bars to each panel of Figure S1B

We thank the editor for the suggestion. Scale bars are now added to individual panel of Figure S1B as suggested.

-please take a look at our Manuscript Preparation guidelines and order your manuscript sections accordingly

-please add a callout to Figure S2C

We thank the editor for pointing this out. In the previous version, the call-out for Figure S2C was mistaken as Fig. S1C. This now has been corrected (L166).

-please provide your manuscript in editable doc format
We now uploaded the manuscript in microsoft word format.

-please provide your tables in editable doc or excel format
We now uploaded the tables in microsoft word format.

-please use the [10 author names, et al.] format in your references (i.e. limit the author names to the first 10)

The references are now updated to match Life Science Alliance requirement (10 author names, *et al.*) and in-text citations as (Authors *et al.*, Year).

Reviewer #2 (Comments to the Authors (Required)):

I am happy with the revisions made by the authors. There are some points to be addressed.

Lines 381-383. This reads speculative and subjective. Could the authors rephrase this point by giving some details for what remains to be shown in Ref 45 and this study?

We thank the reviewer for the suggestion. We now rephrased this point (L395-401) to be less speculative and subjective, as well as suggesting what can be considered in future work (examine the dose-dependent effect of M β CD on membrane integrity, RBOHD behaviors and ROS production).

Lines 405-410. This logic nicely explains MAPK activation being decoupled from DSF effects on CME and receptor internalization. However, a ROS burst is typically induced earlier, and is not well explained. To comprehend all data, it seems better to predict the existence of another as-yet-identified step influencing PRR-NADPH oxidases, rather than simply attributing to time differences between their first detection points.

We thank the reviewer for the comment. As the reviewers also suggested in the previous round of reviewing, we do not exclude other unknown effect that DSF may also exert on other PRRs/NADPH oxidases behaviors as possible explanation, beside the differences in detection points of different assays. This was mentioned in our discussion section (line 383-393).

In Fig 6. FLS2 looks to be homo-dimerized. As discussed with the reviewers, the data available are not strong enough for the occurrence of FLS2 homo-dimerization.

We thank the reviewer for the constructive suggestion. We want to note that in the Figure 6 legend and manuscript main text, we have now replaced the concept of homo-dimerization of FLS2 with nano-clustering as Reviewer 1 previously suggested. We have modified the model in Figure 6 and now shifted the FLS2 molecules to be in close proximity with different nano-clustering states of the molecules and to avoid interpreting this as homo-dimerization (though at the same time, we also want to note that our homoFRET results do suggest that heterogeneity of FLS2 oligomerization also exist although we cannot explicitly measure their clustering or oligomerization states).

July 30, 2020

RE: Life Science Alliance Manuscript #LSA-2020-00720RR

Prof. Yansong Miao
School of Biological Sciences, Nanyang Technological University, Singapore 637551, Singapore
School of Chemical and Biomedical Engineering, Nanyang Technological University, Singapore
637459, Singapore
60 Nanyang Drive
Singapore 637551
Singapore

Dear Dr. Miao,

Thank you for submitting your Research Article entitled "The bacterial QS signal DSF hijacks Arabidopsis sterol biosynthesis to suppress plant immunity". It is a pleasure to let you know that your manuscript is now accepted for publication in Life Science Alliance. Congratulations on this interesting work.

DISTRIBUTION OF MATERIALS:

Again, congratulations on a very nice paper. I hope you found the review process to be constructive and are pleased with how the manuscript was handled editorially. We look forward to future exciting

submissions from your lab.

Sincerely,

Reilly Lorenz
Editorial Office Life Science Alliance
Meyrhofstr. 1
69117 Heidelberg, Germany
t +49 6221 8891 414
e contact@life-science-alliance.org
www.life-science-alliance.org